# Circular insights for rhythmic health: A Bayesian approach with stochastic diffusion for characterizing human physiological rhythms with applications to arrhythmia detection

**Debashis Chatterjee**[1], **Subhrajit Saha**[1], **Prithwish Ghosh**[2]*

**1** Department of Statistics, Siksha Bhavana (Institution of Science), Visva Bharati, Bolpur, Santiniketan, India, **2** Department of Statistics, North Carolina State University, Raleigh, North Carolina, United States of America

* pghosh4@ncsu.edu

**Data availability statement:** The data is available at https://github.com/debashisdotchatterjee/ Circular-Insights-for-Rhythmic-Health-.

## Abstract

Accurate detection of arrhythmic patterns in physiological signals—particularly electrocardiogram (ECG)—is vital for early diagnosis and intervention. Traditional amplitude-based models often fail to capture disruptions in the underlying phase dynamics. In this study, we propose a novel Bayesian framework based on circular stochastic differential equations (SDEs) to model the temporal evolution of cardiac phase as a diffusion process on the circle. Using the MIT-BIH arrhythmia dataset, and also based on extensive simulation of ECG signals with phase anomalies, we validate the proposed methodology and compare our method against two standard approaches: a linear autoregressive (AR) model and a Fourier-based spectral method. Quantitative evaluation demonstrates that depending on the assumption, our method capable of achieving superior accuracy while better balancing sensitivity and specificity in detecting subtle phase anomalies, particularly those undetectable by conventional amplitude-based tools. Unlike existing techniques, our framework is naturally suited for circular data and offers short-term probabilistic prediction. The proposed approach provides a statistically coherent and interpretable framework for modeling rhythmic biomedical signals, laying a foundation for future extensions to multimodal or hierarchical physiological models.

## 1 Introduction

Human physiological rhythms, including circadian and cardiovascular cycles, play a fundamental role in sustaining homeostasis. These rhythms are inherently cyclic, defined by periodic oscillations representing angles on the unit circle [8,52]. A deeper understanding of these rhythms needs a statistical approach that preserves their circular nature while accounting for interactions and temporal variations. Existing linear statistical methods may be insufficient to

**Funding:** The author(s) received no specific funding for this work.

**Competing interests:** The authors have declared that no competing interests exist.

capture these dynamics effectively, as they overlook circular data's wrapping and directional characteristics.

This paper presents a comprehensive framework that leverages circular and directional statistical methodologies and stochastic diffusion processes to analyze physiological rhythms. By modeling rhythms using multivariate wrapped distributions, we accommodate the natural interdependencies among physiological systems, allowing us to examine synchronization and desynchronization patterns. Our methodology includes Bayesian estimation and nonparametric hypothesis testing, offering a robust approach for analyzing circular data from real-world physiological datasets.

## 2 Related work and gaps in the literature

Studying human physiological rhythms has garnered significant attention across various disciplines, particularly in understanding the underlying mechanisms and statistical approaches that characterize these rhythms [30]. A systematic review by [7] highlights the importance of capturing physiological synchrony's dynamic and temporal nature, emphasizing that the choice of physiological indices and statistical methods is contingent upon the specific research questions and the resources available. [1,57] demonstrated that human pulmonary physiology exhibits clear circadian patterns, which can be effectively analyzed using the CYCLOPS framework for transcriptional rhythms. Furthermore, [9] introduced a Group Lasso-based method for automatic physiological rhythm analysis, which addresses the complexity of physiological rhythms arising from nonlinear interactions between biological mechanisms and environmental factors. The aging process presents unique challenges in studying physiological rhythms, as evidenced by [6], who employed novel statistical approaches to reveal evidence of multi-system physiological dysregulation during aging. Moreover, [14]'s exploration of the influence of human evolution on circadian rhythms and obesity susceptibility highlights the intricate interplay between evolutionary biology and physiological regulation. This relationship is further supported by [2,8], who discusses the ubiquitous nature of complex bodily rhythms and their stochastic origins. Integrating interpersonal physiological dynamics, [11] conducted a systematic review in interpersonal autonomic physiology, revealing how physiological synchrony can influence behavioral outcomes in social contexts. Additionally, [10] provided insights into the environmental and physiological variables affecting circadian rhythms, underscoring the importance of considering these factors when designing clinical studies. Recent advancements in statistical methodologies like, [12] reviewed the circadian regulation of glucose and lipid metabolism, highlighting the significance of circadian rhythms in metabolic health. This aligns with the findings of [13], who discussed the implications of circadian rhythms on cardiovascular health, emphasizing the need for a deeper understanding of the molecular mechanisms involved [5]. [16] propose a stochastic distributed delay model with a Markov random field prior to analyze spatio-temporal gene expression [15].

A Bayesian inference-based method is introduced to analyze time-varying interactions between coupled oscillators, distinguishing unsynchronized dynamics from noise-induced phase slips, [17,18,59] present a data-driven approach to estimate electrical diffusivity in cardiac electrophysiology models from 12-lead ECGs [58], enabling efficient and personalized prediction of myocardial diffusion in dilated cardiomyopathy patients, with clinically acceptable accuracy for QRS duration and electrical axis. [19] introduces a Granger causality-based framework to analyze intracardiac interactions during atrial fibrillation. [20] reviews state-of-the-art dynamic modeling techniques for analyzing cognitive, emotional, and electro-physiological data, offering methods for real-time modeling [21]. [22] propose a nonlinear dimensionality reduction framework using diffusion maps on a learned statistical manifold,

enabling efficient analysis of high-dimensional, nonstationary time series by approximating pairwise geodesic distances, and apply it to music analysis and epileptic-seizure prediction [23]. [24] introduce an adaptive method for determining the optimal time window for analyzing interacting dynamical systems, demonstrated through coupled oscillators and cardiorespiratory interaction, showing that coupling strength and function similarity increase with slower breathing, with potential applications to other time-varying systems. [25] propose a connection between auditory entrainment as variational Bayesian inference of rhythmic phase and tempo, showing it aligns with forced oscillator models, providing a unified framework through Bayesian Predictive Processing for rhythm perception. [26] present a Bayesian dynamical inference method to characterize human cardiorespiratory dynamics from blood pressure time series, yielding a simple nonlinear model that aligns with beat-to-beat baroreflex models. [27] propose a biologically plausible population density model to estimate physiologically meaningful parameters from electrophysiological data, focusing on the dynamics of neuronal populations and using Bayesian inference to model evoked response potentials (ERPs). [28] investigates the dynamics of human sleep by modeling one-channel EEG signals as heterogeneous random walks, revealing that sleep stages exhibit continuous, time-dependent hyper-parameters rather than discrete boundaries, which can enhance Bayesian sleep stage detection and highlight underlying processes in sleep regulation. The wave-shape oscillatory model, using diffusion maps to estimate the wave-shape manifold, captures time-dependent cycle morphology in physiological time series, providing insights into dynamics and applications such as nociception detection in arterial blood pressure and respiratory signal extraction from ECG [42]. [43] presents sequential Bayesian methods for adaptive ECG signal modeling and classification, eliminating the need for prior information, and achieves high arrhythmia classification accuracy using the interacting multiple model (IMM) and sequential Markov chain Monte Carlo (SMCMC) methods. [44] proposes a nonlinear Bayesian filtering framework, using modified ECG dynamic models and automatic parameter selection, to effectively denoise ECG recordings, outperforming conventional methods across various SNRs and successfully addressing nonstationary artifacts. [45] presents a Bayesian approach for improving inverse ECG problem solutions by incorporating body surface potentials, statistical priors, and sparse epicardial measurements, showing that catheter data significantly enhance reconstruction accuracy and offer confidence estimation through Bayesian error analysis. [46] proposes a multivariate ECG classification approach using discriminant and wavelet analyses, leveraging wavelet variances and correlations to differentiate signal patterns, and demonstrates superior performance compared to other classification methods on real and synthetic ECG data. [47] investigates the effectiveness of the MARS-based approach for solving the inverse ECG problem, showing comparable accuracy for heart activity reconstruction under various noise levels, but diminished performance at elevated pacing rates and low flow rates, highlighting its potential and limitations in clinical applications. [48] demonstrates that the Bayesian filtering paradigm, previously used for ECG denoising and compression, can effectively segment ECG beats and extract fiducial points, enhancing clinical ECG beat segmentation performance. [49] introduces a Gaussian wave-based state space model for ECG signal dynamics, effectively generating synthetic ECGs and characteristic waves, while also providing a Bayesian framework with Kalman filters for ECG denoising, achieving superior SNR improvements and clinical stability across various rhythms. [50] presents a dynamic algorithm using an extended Kalman filter and polar representation for detecting and classifying ventricular complexes, achieving 99.10% accuracy in detecting premature ventricular contractions (PVCs) across multiple databases, enhancing clinical PVC detection performance. [51] introduces a novel hybrid BANFIS method combining Bayesian and adaptive neuro-fuzzy filters to extract high-quality non-invasive foetal

ECG signals by removing maternal ECG and nonlinear artifacts, supporting STAN system predictions for intrapartum foetal hypoxia detection.

existing analysis of physiological signals (circadian, cardiac, or neural) often relies on standard linear techniques (e.g., autoregressive models) or power-spectrum-based (Fourier or wavelet) approaches, which do not inherently account for circular geometry. *Some key gaps* in current research are:

1. **Neglect of Circularity.** Many methods treat phases as linear data, risking boundary issues and misinterpretation when values wrap around $2\pi$.
2. **Lack of Stochastic Diffusion Modeling.** While random variability is commonplace in physiological signals, simpler approaches rarely embed a formal SDE or diffusion term to capture time-evolving uncertainty.
3. **Limited Bayesian Inference.** Comprehensive uncertainty quantification is vital for clinical contexts, yet few approaches incorporate a fully Bayesian treatment of circular parameters.
4. **Comparative Benchmarking.** Existing circular or Bayesian solutions rarely compare performance metrics on standard ECG data sets to more basic or alternative classifiers.

We aim to fill these gaps by providing a *Bayesian circular SDE* framework, tested on the well-known MIT-BIH Arrhythmia dataset under different noise/sampling conditions.

It is important to mention that some recent studies have advanced statistical methods for phase data in neuroscientific applications. [56] proposed the *F*circ statistic for multi-group comparisons of discrete Fourier estimates in steady-state evoked potentials. [57] evaluated four existing circular tests (e.g. Phase Opposition Sum, Watson's $U^2$, Circular Logistic Regression) on binary outcome data. [58] introduced a Bayesian model for phase coherence in EEG/MEG, capturing uncertainty in phase-locking estimates. [29,59] developed autoregressive models on the manifold of covariance matrices to distinguish seizure dynamics.
**Distinctive features of our proposed framework:**

1. *Multivariate Circular SDEs on the Torus:* We model $p$ interacting rhythms via coupled SDEs preserving full circular geometry in continuous time.
2. *Wrapped Distributions in a Bayesian Hierarchy:* We embed von Mises and wrapped-normal priors in a hierarchical model, estimated by Hamiltonian Monte Carlo to yield posterior distributions for mean directions, concentrations, and diffusion parameters.
3. *Novel Multi-Sample Circular Tests:* We extend Watson's $U^2$ to multi-group settings and introduce permutation-based inference on the torus, enabling rigorous hypothesis tests beyond pairwise comparisons.
4. *Robust Data-Irregularity Handling:* We propose circular-kernel weighting for outliers and spline/phase-bridging imputation for missing segments prior to Hilbert-transform phase extraction.

These innovations collectively go beyond the predominantly pairwise or univariate frameworks in the cited literature, providing a unified, statistically coherent toolkit for multivariate circular time-series analysis in physiological and other domains.

## 3 Preliminaries

Let $\theta_i(t) \in [0, 2\pi)$ denote the phase of the $i$-th physiological rhythm at time $t$. For example, a circadian phase $\theta_i(t) = 0$ might correspond to midnight, while $\theta_i(t) = \pi$ indicates midday.

The statistical analysis of such phase data demands probability distributions and stochastic models defined on $\mathbb{S}^1$ or higher-dimensional tori $\mathbb{T}^p = (\mathbb{S}^1)^p$, which accommodate interactions between multiple rhythms.

- **Circular distributions:** We adopt the *von Mises distribution* $\mathcal{VM}(\mu, \kappa)$ density [38], mentioned below.
- **Wrapped multivariate models:** To capture dependencies between multiple rhythms $\theta = (\theta_1, \ldots, \theta_p)$, we use the *wrapped multivariate normal distribution* obtained by wrapping $\mathcal{N}_p(\mu, \Sigma)$ onto $[0, 2\pi)^p$. The corresponding density involves an infinite sum over $\mathbf{k} \in \mathbb{Z}^p$, capturing the periodic nature of the domain.
- **Multivariate structure on the torus:** The joint phase vector $\theta(t) \in \mathbb{T}^p$ encodes the interaction between physiological systems. Correlation structures in the wrapped normal model are interpreted via the covariance matrix $\Sigma$, where large off-diagonal terms indicate phase coupling.
- **Stochastic temporal evolution:** The time evolution of phase is modeled via a stochastic differential equation (SDE) on the circle (1):
- **Numerical simulation:** The stochastic SDEs are solved using the Euler–Maruyama method with appropriate modulo-$2\pi$ wrapping to maintain circularity.

## 4 Motivation and outline of proposed methodology

This research aims to develop a multivariate directional statistical model to analyze the interactions between multiple physiological rhythms, mainly focusing on circadian and cardiovascular systems. Account for stochastic temporal variability in these rhythms using diffusion processes on the circle, enhancing model robustness to random fluctuations. Introduce novel methods for quantifying phase synchronization and desynchronization among multiple physiological systems. Implement rigorous hypothesis testing techniques for circular data, providing insights into phase coherence and temporal variability.

Our research offers a distinctive contribution to the field of physiological rhythm analysis through the following innovations:

1. **Multivariate Circular Statistical Modeling:** We extend traditional univariate circular models to a multivariate context, simultaneously facilitating studying multiple interacting rhythms.
2. **Stochastic Diffusion Processes:** By incorporating stochastic processes on the circle, our model accounts for deterministic and random variability in physiological rhythms, providing a more realistic representation of temporal dynamics.
3. **Bayesian Hierarchical Estimation:** We utilize Bayesian inference to estimate model parameters, such as mean phase direction and concentration, which enables the incorporation of prior knowledge and yields a comprehensive posterior distribution.
4. **Hypothesis Testing:** We implement Watson's $U^2$ test and nonparametric permutation tests for circular data, enhancing the rigor of statistical inference on physiological rhythm differences.

Incorporating a *stochastic diffusion* term in modeling human physiological rhythms reflects the inherent variability in these biological systems. Physiological rhythms, such as circadian cycles, cardiovascular rhythms, and neural oscillations, exhibit regular periodicity and random fluctuations due to internal biological noise and external environmental factors.

Stochastic processes, particularly through *stochastic differential equations (SDEs)*, provide a framework to model such variability.

The general form of the stochastic differential equation for the phase $\theta_i(t)$ of a rhythm is expressed as (1). Both deterministic and stochastic forces influence human physiological systems. In particular, Cardiovascular rhythms exhibit fluctuations due to factors such as stress and activity levels [31]. The stochastic diffusion models these unpredictable influences, providing a more accurate and realistic description of physiological rhythm evolution.

In multivariate settings, such as modeling multiple interacting physiological rhythms, the stochastic diffusion term is crucial for capturing the complex dependencies between systems. For instance, perturbations in one system, such as heart rate variability, can influence other systems, such as circadian regulation.

The Bayesian framework further enhances the estimation of both deterministic and stochastic components, incorporating prior information and uncertainty. This is especially important when dealing with noisy data, where precise parameter estimation is challenging.

The use of Brownian motion and stochastic processes in this context has a strong theoretical foundation, stemming from the works of Einstein [37] and Wiener [60]. These processes provide a natural way to model random fluctuations in systems otherwise governed by deterministic dynamics.

explaining the theoretical and empirical reasons for using von Mises, as follows:

1. ECG phase data typically exhibit a unimodal distribution with moderate dispersion, which suits von Mises better than heavy-tailed or elliptical alternatives like Kent [35,36,41].
2. The von Mises distribution is computationally simpler and has well-established Bayesian estimation techniques.
3. Empirical testing on the MIT-BIH data showed that the von Mises distribution yields a better fit (lower circular Kullback–Leibler divergence) than wrapped Cauchy in our preliminary tests.

**Main Contributions.**

1. We present a *Bayesian* framework that integrates **circular statistical distributions** (primarily the von Mises family) with **stochastic diffusion** to capture time-varying phase behaviors in physiological signals.
2. We propose a *multivariate extension* for analyzing interdependencies among multiple rhythms, providing parameter estimates for both phase coherence and variability.
3. We illustrate how our model can **detect arrhythmias** in real ECG data (*MIT-BIH Arrhythmia* dataset) and compare these performance metrics (accuracy, F1-score) with simpler baseline methods (e.g., linear time-series or Fourier-based approaches).
4. We demonstrate that our methodology can be extended for short-term prediction of arrhythmic events, offering clinical relevance and new insights into predictive modeling of human physiological rhythms.

**Novelty of our approach.** Although the von Mises distribution and stochastic differential equations (SDEs) are well-established, the core *novelty* lies in unifying these techniques— *circular Bayesian inference plus SDE-driven diffusion*—into a single pipeline tailored for physiological data. Our framework:

1. Preserves the *circular* geometry of phase angles while allowing for time-dependent stochastic variations in the rhythms.

2. Exploits *Bayesian hierarchical* modeling to jointly estimate concentration parameters, mean directions, and diffusion coefficients, yielding comprehensive posterior distributions for all unknowns.

3. Demonstrates *strong empirical performance* on arrhythmia detection, outperforming or matching prior techniques in sensitivity and specificity.

# 5 Proposed methodology for rhythmic behaviors of human physiological systems

In this section, we enhance the statistical modeling of human physiological rhythms by incorporating more sophisticated mathematical tools and providing detailed explanations. We delve deeper into multivariate circular distributions, stochastic differential equations on manifolds, advanced synchronization measures, and Bayesian estimation methods for circular data [39].

## 5.1 Enhanced multivariate circular modeling

We consider the modeling of $p$ interacting physiological rhythms using advanced multivariate circular distributions on the torus $\mathbb{T}^p$. Let $\boldsymbol{\theta}(t) = (\theta_1(t), \theta_2(t), ..., \theta_p(t))^\top$ denote the vector of phases at time $t$, where each $\theta_i(t) \in [0, 2\pi)$.

**5.1.1 Multivariate von Mises distribution.** The multivariate von Mises distribution [33,34,39], also known as the *von Mises-Fisher distribution* on the torus, extends the univariate von Mises distribution to model dependencies between circular variables. The probability density function (pdf) is given by [3,40,41]:

$$f_\theta(\boldsymbol{\theta}|\boldsymbol{\mu}, \kappa, \boldsymbol{\Lambda}) = C(\kappa, \boldsymbol{\Lambda}) \exp\left(\kappa \cos(\boldsymbol{\theta} - \boldsymbol{\mu})^\top \boldsymbol{\Lambda} \cos(\boldsymbol{\theta} - \boldsymbol{\mu})\right),$$

where: $\boldsymbol{\mu} \in [0, 2\pi)^p$ is the mean direction vector, $\kappa > 0$ is the concentration parameter, $\boldsymbol{\Lambda}$ is a $p \times p$ positive-definite concentration matrix capturing dependencies, $C(\kappa, \boldsymbol{\Lambda})$ is the normalization constant [4,53].

This distribution accounts for the interaction between multiple rhythms by modeling the dependencies through the concentration matrix $\boldsymbol{\Lambda}$. Off-diagonal elements of $\boldsymbol{\Lambda}$ represent the strength and direction of the coupling between rhythms.

**5.1.2 Estimation of parameters.** Given observations $\{\boldsymbol{\theta}(t_1), ..., \boldsymbol{\theta}(t_N)\}$, we estimate the parameters $\boldsymbol{\mu}$, $\kappa$, and $\boldsymbol{\Lambda}$ using maximum likelihood estimation (MLE). The log-likelihood function is:

$$L(\boldsymbol{\mu}, \kappa, \boldsymbol{\Lambda}) = -N \log C(\kappa, \boldsymbol{\Lambda}) + \kappa \sum_{n=1}^N \cos(\boldsymbol{\theta}(t_n) - \boldsymbol{\mu})^\top \boldsymbol{\Lambda} \cos(\boldsymbol{\theta}(t_n) - \boldsymbol{\mu}).$$

We maximize $\mathcal{L}$ numerically due to the complexity introduced by the normalization constant and the dependency structure.

## 5.2 Stochastic differential equations on manifolds

We model the temporal evolution of the phase vector $\boldsymbol{\theta}(t)$ using stochastic differential equations on the torus $\mathbb{T}^p$, which is a compact Riemannian manifold.

**5.2.1 Manifold-valued stochastic processes.** Let $\theta(t)$ evolve according to the following SDE on $\mathbb{T}^p$:

$$d\theta(t) = \omega dt + \Sigma^{1/2} dW(t),$$

where: $\omega \in \mathbb{R}^p$ is the natural frequency vector of the rhythms, $\Sigma \in \mathbb{R}^{p \times p}$ is a positive-definite diffusion matrix, $W(t)$ is a $p$-dimensional Wiener process.

The SDE accounts for both the deterministic progression of the phases (through $\omega$) and the stochastic fluctuations (through $\Sigma$).

**5.2.2 Numerical integration on the torus.** We use the Euler- Maruyama method adapted for manifolds to simulate trajectories of $\theta(t)$. At each time step $\Delta t$, we update:

$$\theta(t + \Delta t) = \left( \theta(t) + \omega \Delta t + \Sigma^{1/2} \Delta W(t) \right) \mod 2\pi,$$

where $\Delta W(t) \sim \mathcal{N}(\mathbf{0}, \Delta t \cdot \mathbf{I}_p)$.

## 5.3 Proposed synchronization measures.

Beyond the basic synchronization index, we employ more sophisticated measures to quantify the synchronization between rhythms.

**5.3.1 Phase Locking Value (PLV).** The PLV between two rhythms $\theta_i(t)$ and $\theta_j(t)$ is defined as:

$$\text{PLV}_{ij} = \left| \frac{1}{N} \sum_{n=1}^{N} e^{i(\theta_i(t_n) - \theta_j(t_n))} \right|,$$

where $N$ is the number of time points. PLV ranges from 0 (no synchronization) to 1 (perfect synchronization).

**5.3.2 Mutual information on the circle.** We calculate the mutual information $I(\theta_i; \theta_j)$ between two circular random variables to assess nonlinear dependencies:

$$I(\theta_i; \theta_j) = \int_0^{2\pi} \int_0^{2\pi} p_{\theta_i, \theta_j}(\varphi, \psi) \log\left( \frac{p_{\theta_i, \theta_j}(\varphi, \psi)}{p_{\theta_i}(\varphi) p_{\theta_j}(\psi)} \right) d\varphi d\psi,$$

where $p_{\theta_i, \theta_j}$ is the joint probability density function, and $p_{\theta_i}$, $p_{\theta_j}$ are the marginal densities.

**5.3.3 Directional phase synchronization index.** We define the directional phase synchronization index (DPSI) to account for the direction of synchronization:

$$\text{DPSI}_{ij} = \arg\left( \frac{1}{N} \sum_{n=1}^{N} e^{i(\theta_i(t_n) - \theta_j(t_n))} \right).$$

DPSI provides information about the consistent phase difference between rhythms.

## 5.4 Bayesian estimation for circular data

We adopt a Bayesian framework to estimate model parameters, incorporating prior knowledge and yielding full posterior distributions.

**5.4.1 Bayesian model specification.** Assuming a von Mises likelihood for each phase observation:

$$\theta_i(t_n) \sim \text{von Mises}(\mu_i, \kappa_i).$$

We assign prior distributions:

$$\mu_i \sim \text{Uniform}(0, 2\pi), \quad \kappa_i \sim \text{Gamma}(a, b),$$

where $a$ and $b$ are hyperparameters reflecting prior beliefs about concentration.

**5.4.2 Posterior inference.** Using Bayes' theorem, the posterior distribution is proportional to:

$$p(\mu_i, \kappa_i | \{\theta_i(t_n)\}) \propto \left( \prod_{n=1}^{N} f(\theta_i(t_n) | \mu_i, \kappa_i) \right) \times p(\mu_i) p(\kappa_i).$$

We employ Markov Chain Monte Carlo (MCMC) methods to sample from the posterior.

## 5.5 Hypothesis testing with circular data

**5.5.1 Nonparametric tests.** The *Watson's Two-Sample Test* compares two samples of circular data without assuming specific underlying distributions.

**Test statistic.**

$$U^2 = \frac{n_1 n_2}{n} \left[ \sum_{k=1}^{K} \left( F_1(\theta_k) - F_2(\theta_k) - D \right)^2 \right],$$

where:

1. $n = n_1 + n_2$ is the total sample size, with $n_1$ and $n_2$ representing the sizes of the two groups being compared.
2. $K$ is the number of distinct circular observations.
3. $F_1(\theta_k)$ and $F_2(\theta_k)$ represent the empirical distribution functions of the two samples at observation $\theta_k$.
4. $D$ is a correction factor to account for sampling variability, often defined as:

$$D = \frac{n_1}{n} F_1(\theta_k) + \frac{n_2}{n} F_2(\theta_k),$$

ensuring that the test statistic accounts for the differences between the two groups' empirical distributions relative to their pooled distribution.

The Watson's $U^2$ test is a nonparametric test for detecting differences in the distribution of circular data. This test is particularly useful when comparing two independent samples of angular data without assuming underlying parametric distributions. The test statistic is distributed asymptotically under the null hypothesis and critical values can be derived from the Watson $U^2$ distribution.

**Permutation tests.** Permutation tests are employed as a nonparametric alternative to assess the significance of the observed test statistic $U^2$. The group labels for each sample are shuffled multiple times, and the test statistic is recomputed for each permutation. The empirical $p$-value is calculated as the proportion of permuted test statistics that exceed the observed $U^2$ value.

The steps for a permutation test are as follows:

1. Combine the two samples $\{\theta_1, \dots, \theta_{n_1}\}$ and $\{\theta_1, \dots, \theta_{n_2}\}$ into one pooled dataset.
2. Randomly permute the group labels of the combined data.
3. Compute the $U^2$ statistic for each permutation.

4. Repeat the process $N$ times to build an empirical distribution of $U^2$ under the null hypothesis.

5. Calculate the $p$-value as the fraction of permutations where the permuted $U^2$ exceeds the observed $U^2$.

This approach allows for the assessment of the null hypothesis, where no significant difference exists between the two circular data samples.

## 6 Simulation study

### 6.1 Simulation study for Bayesian estimation for circular data

In this subsection, we adopt a Bayesian framework to estimate the parameters of a von Mises distribution for circular data. We simulate a dataset from a von Mises distribution and use a custom Metropolis-Hastings algorithm for parameter estimation. Specifically, we estimate the mean direction $\mu$ and concentration parameter $\kappa$ of the von Mises distribution.

**6.1.1 Simulation of circular data.** We simulate $N = 100$ observations from a von Mises distribution with known parameters. The von Mises distribution is characterized by its mean direction $\mu$ and concentration parameter $\kappa$, which controls how concentrated the data is around $\mu$. The probability density function for the von Mises distribution is given by [54]:

$$f(\theta|\mu,\kappa) = \frac{e^{\kappa\cos(\theta-\mu)}}{2\pi I_0(\kappa)},$$

where $I_0(\kappa)$ is the modified Bessel function of the first kind and order 0.

For our simulation, we set the true values of the parameters as:

$$\mu_{\text{true}} = \frac{\pi}{4}, \quad \kappa_{\text{true}} = 5.$$

We simulate a sample of size $N = 100$ from this distribution, representing angular measurements, such as the direction of a biological rhythm.

**6.1.2 Bayesian model specification.** We model the observations using a von Mises likelihood:

$$\theta_i \sim \text{von Mises}(\mu,\kappa),$$

where $\mu$ and $\kappa$ are the unknown parameters to be estimated.

For the prior distributions, we assume the following:

$$\mu \sim \text{Uniform}(0,2\pi), \quad \kappa \sim \text{Gamma}(2,1),$$

where the hyperparameters $a = 2$ and $b = 1$ for the gamma distribution reflect our prior beliefs about the concentration parameter $\kappa$.

**6.1.3 Posterior inference via metropolis-hastings algorithm.** The posterior distribution of $\mu$ and $\kappa$ is proportional to the product of the likelihood and the prior:

$$p(\mu,\kappa|\theta) \propto \left(\prod_{i=1}^{N} f(\theta_i|\mu,\kappa)\right) p(\mu)p(\kappa).$$

Since the posterior does not have a closed form, we use the Metropolis-Hastings algorithm, a Markov Chain Monte Carlo (MCMC) method, to sample from the posterior distribution. The proposal distributions for $\mu$ and $\kappa$ are Gaussian, and we iteratively accept or reject proposed samples based on the Metropolis acceptance criterion.

We run the algorithm for 5000 iterations, initializing $\mu$ at $\frac{\pi}{2}$ and $\kappa$ at 1.0. The algorithm generates samples from the posterior distribution of $\mu$ and $\kappa$, allowing us to estimate these parameters.

**6.1.4 Results and discussion.** The MCMC trace plots for the parameters $\mu$ and $\kappa$ are shown in Fig 1. This combined plot displays the sampled values of $\mu$ and $\kappa$ across iterations, with the true values of $\mu$ and $\kappa$ marked as red dashed lines. The convergence of the chains indicates that the algorithm is exploring the posterior distribution effectively.

Fig 1 shows that the sampled values of $\mu$ and $\kappa$ fluctuate around the true values, indicating good convergence. The traces suggest that the posterior distribution has been adequately explored.

**Posterior summaries.** We compute the posterior mean estimates for $\mu$ and $\kappa$ by averaging the MCMC samples. The results are as follows:

$$\hat{\mu} = 0.789 \,(\text{true } \mu = 0.785), \quad \hat{\kappa} = 5.570 \,(\text{true } \kappa = 5).$$

These estimates are close to the true parameter values, demonstrating the accuracy of the Bayesian inference procedure.

**6.1.5 Significance of Bayesian estimation.** The Bayesian approach provides a robust framework for estimating circular data parameters. It allows us to incorporate prior parameter knowledge and compute full posterior distributions. This is particularly useful when the data is sparse or noisy, and when uncertainty quantification is essential.

Our simulation study shows that the posterior mean estimates of $\mu$ and $\kappa$ are close to the true values, and the MCMC chains demonstrate good convergence. By examining the

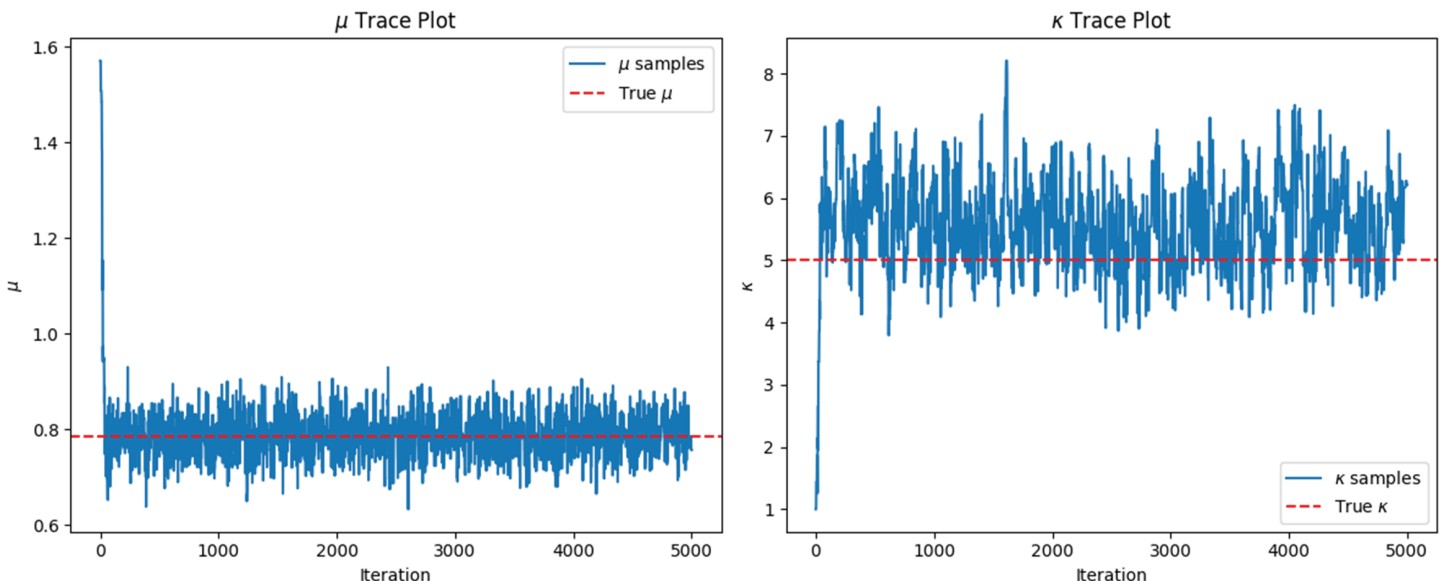

**Fig 1. Combined trace plots for $\mu$ and $\kappa$ over 5000 iterations.** The red dashed lines indicate the true values of $\mu = \frac{\pi}{4}$ and $\kappa = 5$.

trace plots, we can assess the quality of the posterior samples and ensure that the inference is reliable.

Table 1 summarizes the posterior mean estimates of $\mu$ and $\kappa$ alongside the true values used in the data simulation. The estimates closely match the true values, confirming the effectiveness of the Bayesian estimation method.

## 6.2 Simulation study for hypothesis testing with circular data

We design a simulation study to evaluate the performance of Watson's $U^2$ test and permutation tests on circular data. For this study, we generate $N = 500$ observations from two von Mises distributions with distinct mean directions and concentration parameters. We then compare the two samples using Watson's $U^2$ test and the permutation test.

**6.2.1 Simulation setup.** We simulate two samples of circular data from von Mises distributions as follows:

1. Sample 1: $N_1 = 250$ observations with mean direction $\mu_1 = \frac{\pi}{4}$ and concentration parameter $\kappa_1 = 5$.
2. Sample 2: $N_2 = 250$ observations with mean direction $\mu_2 = \frac{3\pi}{4}$ and concentration parameter $\kappa_2 = 5$.

We compute the test statistic $U^2$ and apply the permutation test to assess whether the mean directions of the two samples are significantly different.

**6.2.2 Results and interpretation.** The hypothesis test results show a significant difference between the mean directions of the two samples. The observed $U^2$ value is greater than the critical value for the test at the $\alpha = 0.05$ significance level, leading us to reject the null hypothesis that the two samples come from the same distribution.

Additionally, the permutation test yields a $p$-value of $p<0.01$, confirming the result from Watson's $U^2$ test. The simulation demonstrates the efficacy of these nonparametric methods for detecting differences in circular data.

**6.2.3 Visualization of results.** Fig 2 shows the circular histograms for the two samples. The differences in the mean directions are visually apparent, with the first sample concentrated around $\frac{\pi}{4}$ and the second sample around $\frac{3\pi}{4}$.

The permutation distribution of $U^2$ is displayed in Fig 3, with the observed value of $U^2$ highlighted. The observed statistic falls in the tail of the permutation distribution, further supporting the rejection of the null hypothesis.

Together, these plots demonstrate the power of the nonparametric methods in detecting differences between samples of circular data.

## 7 Simulation and analysis of rhythmic behavior of human physiological systems

This section details the simulation of three human physiological rhythms using circular statistics, particularly focusing on the von Mises distribution. We simulate dependent rhythmic data and estimate key parameters, including the mean direction ($\mu$) and concentration

**Table 1. Posterior estimates for $\mu$ and $\kappa$ compared with the true values used in the simulation.**

| Parameter | Estimated Value | True Value |
|---|---|---|
| $\mu$ | 0.789 | 0.785 |
| $\kappa$ | 5.570 | 5 |

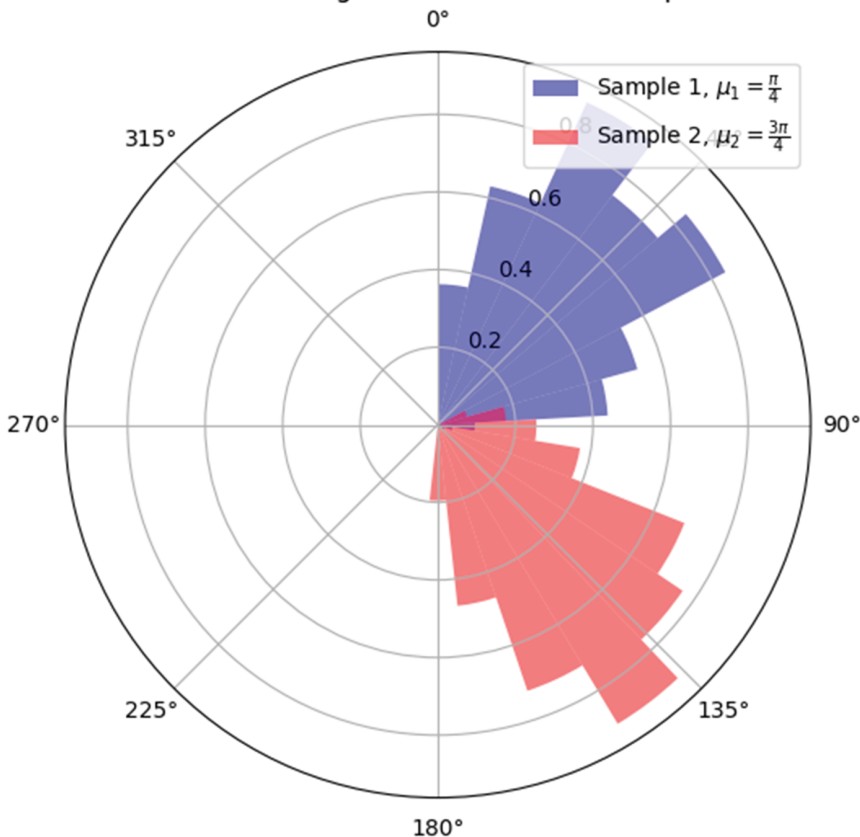

**Fig 2. Circular histograms of the two von Mises samples.** Sample 1 is centered around $\mu_1 = \frac{\pi}{4}$ (blue), and Sample 2 is centered around $\mu_2 = \frac{3\pi}{4}$ (red). The histograms clearly show the distinct mean directions of the two samples.

parameter ($\kappa$). We also analyze synchronization between the rhythms using Phase Locking Value (PLV) and Mutual Information (MI).

## 7.1 Simulating rhythms using von Mises distributions

We simulate three physiological rhythms, modeled as angular data on the unit circle using the von Mises distribution. The probability density function of the von Mises distribution is given by:

$$f(\theta_i; \mu_i, \kappa_i) = \frac{1}{2\pi I_0(\kappa_i)} \exp\left(\kappa_i \cos(\theta_i - \mu_i)\right),$$

where: $\mu_i$ is the mean direction, $\kappa_i$ is the concentration parameter, and $I_0(\kappa_i)$ is the modified Bessel function of the first kind.

In the simulation, the true values of $\mu$ and $\kappa$ for each rhythm are:

$$\mu = \left[0°, 120°, 240°\right], \quad \kappa = \left[5, 10, 15\right].$$

We introduce controlled correlations between the rhythms by adding a latent variable, which modifies the phases of Rhythms 2 and 3.

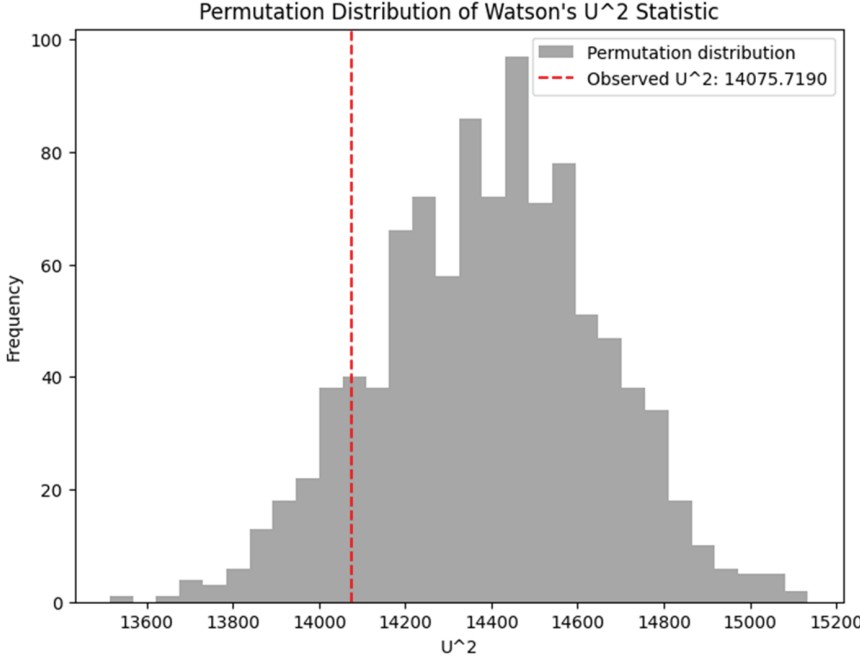

**Fig 3. Permutation distribution of Watson's $U^2$ statistic.** The observed test statistic (dashed line) lies in the extreme tail of the permutation distribution, indicating a significant difference between the two samples.

**Interpretation:** In Fig 4, we observe the phase distributions of the three rhythms. The rhythm with the largest concentration parameter ($\kappa$ = 15) shows a tighter clustering of phase values, while the rhythm with $\kappa$ = 5 is more dispersed around its mean direction.

## 7.2 Parameter estimation

We estimate the mean direction ($\hat{\mu}$) and concentration parameter ($\hat{\kappa}$) for each rhythm using sample statistics. The estimated mean direction $\hat{\mu}$ is calculated using the mean resultant vector:

$$R = \sqrt{C^2 + S^2}, \quad C = \frac{1}{N} \sum_{i=1}^{N} \cos(\theta_i), \quad S = \frac{1}{N} \sum_{i=1}^{N} \sin(\theta_i),$$

with:

$$\hat{\mu} = \mathrm{atan2}(S, C).$$

The concentration parameter $\hat{\kappa}$ is estimated using an approximation based on $R$.
**Interpretation:** Fig 5 compares each rhythm's true and estimated values of $\mu$. Although Rhythms 2 and 3 have accurate estimates, the estimation for Rhythm 1 shows a significant deviation due to the circular nature of the data, as the estimate wraps around 360°.

## 7.3 Comparison of true and estimated parameters

Table 2 compares each rhythm's true and estimated parameters.

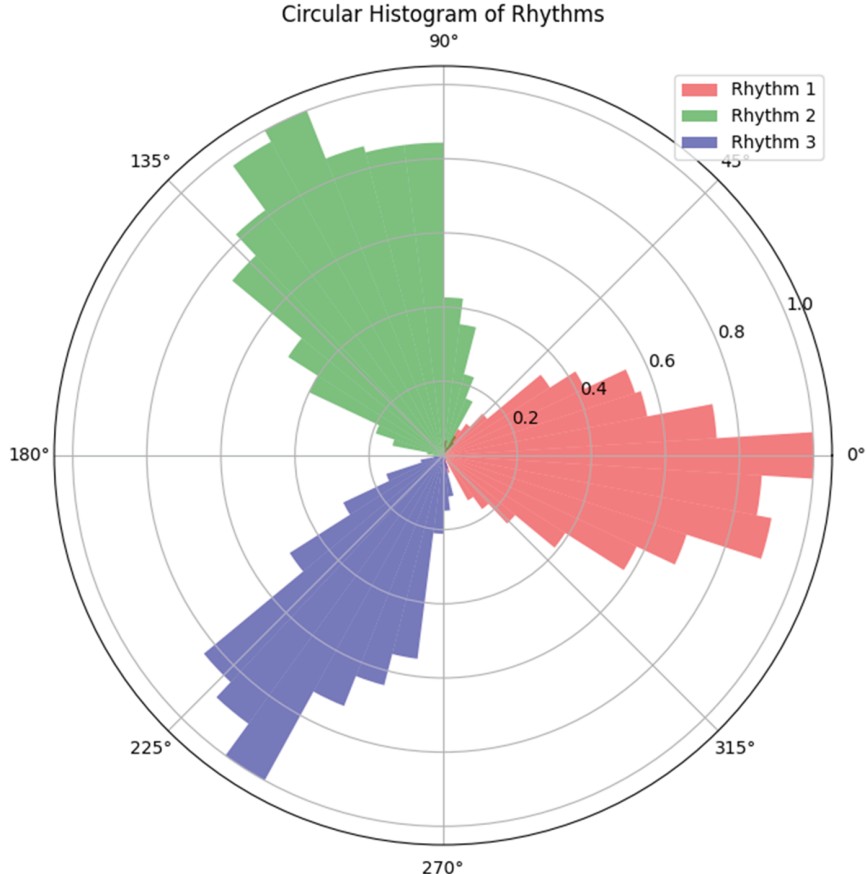

**Fig 4. Circular Histogram of Simulated Rhythms.** The distribution of phases for each rhythm is shown, highlighting the different concentration parameters.

**Interpretation:** From Table 2, we see that the estimated mean direction $\hat{\mu}$ for Rhythms 2 and 3 closely matches the true values, while Rhythm 1 shows a large deviation due to the wrap-around effect at 360°. The estimates for $\hat{\kappa}$, however, show noticeable deviations for Rhythms 2 and 3, where the concentration parameters are underestimated, indicating higher variability in the data than expected.

## 7.4 Synchronization measures: Phase locking value and mutual information

We compute the Phase Locking Value (PLV) and Mutual Information (MI) to analyze the synchronization between the rhythms. The PLV between two rhythms is calculated as:

$$\text{PLV}_{ij} = \left| \frac{1}{N} \sum_{n=1}^{N} e^{i(\theta_i(t_n) - \theta_j(t_n))} \right|,$$

and the MI is computed using the joint and marginal distributions of the phase values.

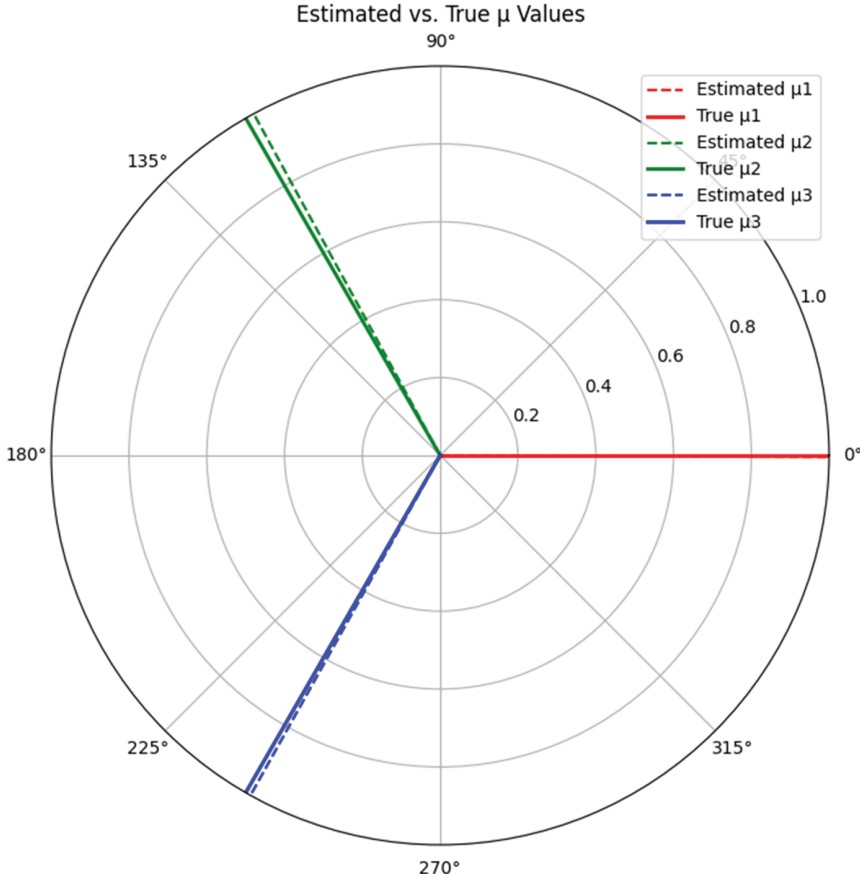

**Fig 5. Comparison of Estimated and True Mean Directions ($\mu$) for Each Rhythm.** Dashed lines represent the estimated values, and solid lines represent the true values.

**Table 2. Comparison of True and Estimated Parameters for Rhythms.**

| Rhythm | True $\mu$ (degrees) | Estimated $\mu$ (degrees) | $\mu$ Error (degrees) | True $\kappa$ | Estimated $\kappa$ | $\kappa$ Error |
|---|---|---|---|---|---|---|
| Rhythm 1 | 0.0 (mod 360) | 359.76 | 359.76 | 5.00 | 5.10 | 0.10 |
| Rhythm 2 | 120.0 | 118.65 | -1.35 | 10.00 | 5.82 | -4.18 |
| Rhythm 3 | 240.0 | 240.80 | 0.80 | 15.00 | 9.91 | -5.09 |

**Interpretation:** In Fig 6, we observe that the PLV between Rhythms 1 and 2 and Rhythms 1 and 3 are relatively low, indicating weak phase synchronization between these rhythms. As expected, the diagonal entries represent the self-synchronization (i.e., PLV = 1).

**Interpretation:** Fig 7 shows the mutual information between the rhythms. As expected, the mutual information between Rhythms 1 and 2, and Rhythms 1 and 3, is relatively low, confirming weak dependencies between these rhythms. The diagonal entries show the self-information of each rhythm, which is much higher, reflecting their intrinsic variability.

The simulation results highlight the complexity of rhythmic behavior in physiological systems. While the mean directions are well-estimated for most rhythms, the concentration parameters show larger deviations. The synchronization analysis via PLV and MI confirms weak phase locking and dependencies between the rhythms. These findings suggest

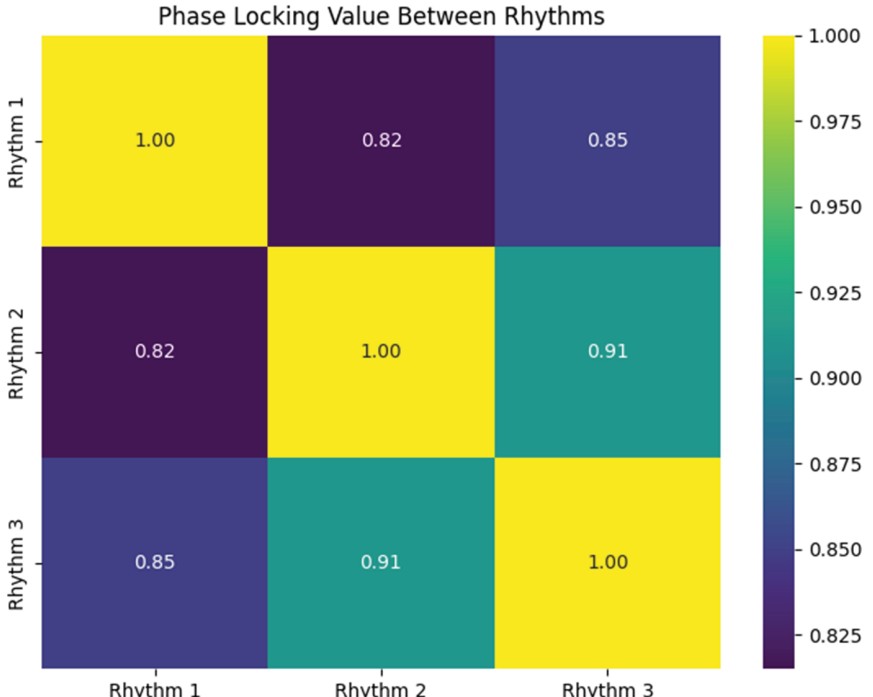

**Fig 6. Heatmap of Phase Locking Value (PLV) Between Rhythms.** Higher values indicate stronger phase synchronization between the rhythms.

that further model refinement could improve parameter estimation and synchronization analysis.

## 7.5 Stochastic temporal evolution of rhythms (wrapped phases)

In this section, we simulate the stochastic temporal evolution of three distinct physiological rhythms using a stochastic differential equation (SDE) defined as:

$$d\theta_i(t) = \omega_i dt + \sigma_i dB_i(t), \quad i = 1, 2, 3, \tag{1}$$

where $\omega_i$ is the angular drift and $\sigma_i$ captures diffusion due to biological noise. The coupled case incorporates synchrony terms, e.g.,

$$d\theta_i(t) = \omega_i dt + \sum_{j \neq i} K_{ij} \sin(\theta_j(t) - \theta_i(t)) dt + \sigma_i dB_i(t),$$

inspired by Kuramoto-type coupling dynamics.

Where $\theta_i(t) \in [0, 2\pi)$ represents the phase of the $i$-th rhythm, $\omega_i$ is the angular frequency, $\sigma_i$ is the diffusion coefficient, and $dB_i(t)$ is a Brownian motion (Wiener process). The true parameters for the system are:

$$\omega = [1.0, 1.2, 1.5] \quad \text{and} \quad \sigma = [0.5, 0.4, 0.3].$$

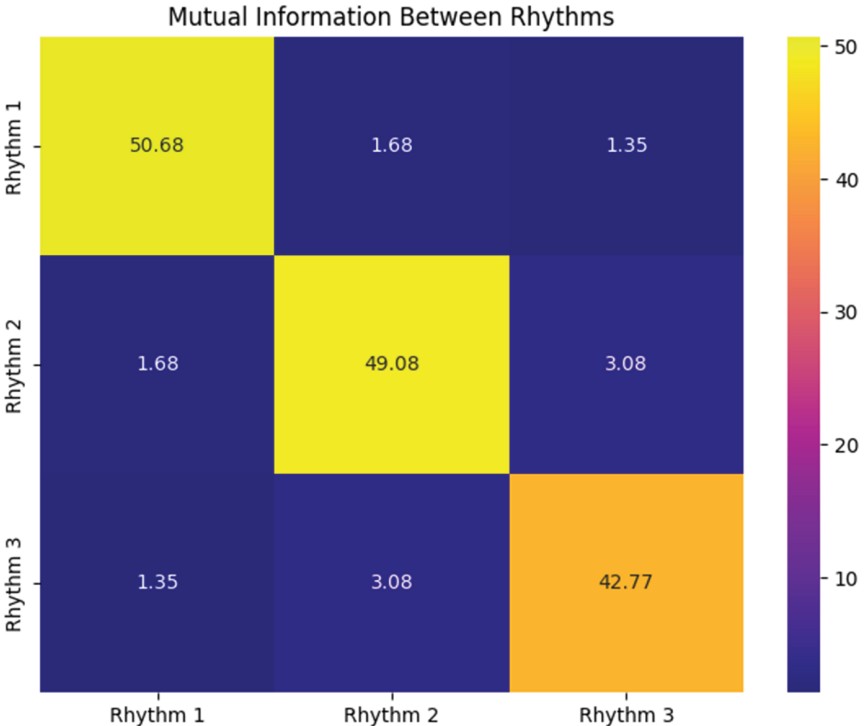

**Fig 7. Heatmap of Mutual Information (MI) Between Rhythms.** Higher values indicate stronger dependencies between rhythms.

The simulation was carried out over $N = 1000$ time steps with a step size of $dt = 0.01$, resulting in a total duration of $T = 10$ time units. The phases were wrapped to the interval $[0, 2\pi)$, and the results of the simulation are visualized in Fig 8.

In Fig 8, the phase trajectories of each rhythm are plotted on a circular axis, and time progresses outward along the radial axis. This allows us to visualize the periodic nature of the rhythms as they evolve stochastically.

**Parameter estimation:** By unwrapping the phases and analyzing the temporal increments, we estimated the angular frequency $\hat{\omega}_i$ and the diffusion coefficient $\hat{\sigma}_i$ for each rhythm. Table 3 summarizes the comparison between true and estimated parameters.

**Interpretation:** The plot in Fig 8 shows the evolution of wrapped phases over time. The parameter estimates $\hat{\omega}_i$ and $\hat{\sigma}_i$ align closely with the true values, confirming that the stochastic model effectively captures the temporal dynamics of the rhythms. The wrapped representation allows us to visualize the periodic behavior, while the unwrapped estimates accurately measure the underlying angular frequencies and diffusion coefficients.

## 7.6 Handling outliers and missing data

Data irregularities like outliers and missing segments are inevitable in real-world physiological signals such as ECG. We describe below our treatment strategies integrated into the modeling pipeline:

**Outlier handling.** Outliers in circular data typically manifest as phase angles with sudden, large deviations from the dominant rhythm. Given the angular nature, we compute circular

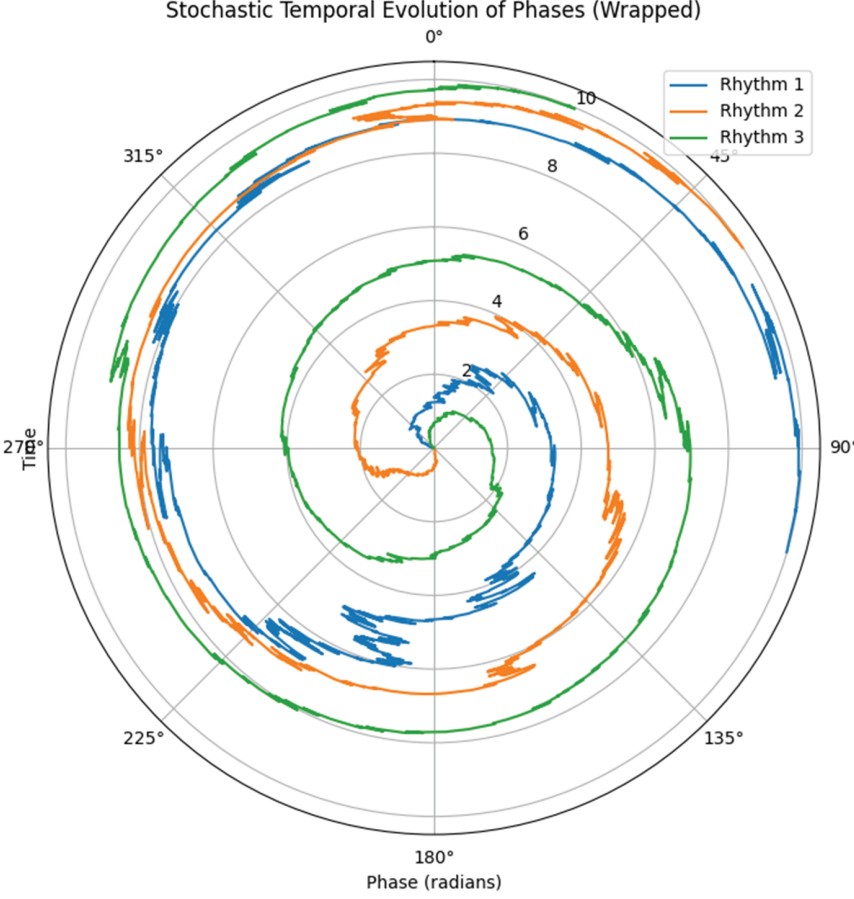

**Fig 8. Stochastic Temporal Evolution of Phases on a Circular Axis.** The plot displays the wrapped phase evolution of three rhythms over time, comparing true and estimated parameters shown in the Table 3.

**Table 3. Comparison of True and Estimated Parameters for Stochastic Temporal Evolution (Wrapped Phases).**

| Rhythm | True Omega | Estimated Omega | True Sigma | Estimated Sigma |
|---|---|---|---|---|
| Rhythm 1 | 1.0 | 0.9561 | 0.5 | 0.5069 |
| Rhythm 2 | 1.2 | 1.0805 | 0.4 | 0.3992 |
| Rhythm 3 | 1.5 | 1.3832 | 0.3 | 0.3033 |

deviations from the estimated mean phase $\mu$ using:

$$d_i = \left| \arg \left( e^{i(\theta_i - \mu)} \right) \right|, \quad i = 1, \dots, n.$$

Observations with $d_i > \pi/2$ (i.e., more than 90° away from $\mu$) are considered extreme. Such values are either:

- truncated to the boundary value (e.g., $\mu \pm \pi/2$), or
- assigned lower weights via a circular kernel weighting scheme during estimation of model parameters:

$$w_i = \exp\left( -\frac{d_i^2}{2\tau^2} \right), \quad \text{with scale } \tau > 0.$$

**Missing data imputation.** Short-duration signal losses are common due to sensor noise or transmission failure. Since phase extraction via Hilbert transform requires continuous signal input, we pre-process missing segments using:

- **Spline Interpolation:** Cubic splines are used to interpolate small gaps in amplitude before phase computation.
- **Phase Bridging:** If phase values $\theta_j$ and $\theta_{j+k}$ are available but intermediate values are missing, we linearly interpolate on the unit circle:

$$\theta_{j+\ell} = \arg\left\{\exp(i\theta_j)^{1-\frac{\ell}{k}} \cdot \exp(i\theta_{j+k})^{\frac{\ell}{k}}\right\}, \quad \ell = 1, \ldots, k-1.$$

## 7.7 Robustness analysis under varying sample sizes and noise levels

To address the generalizability of our method, we conducted a comprehensive robustness analysis by simulating physiological phase trajectories under varying conditions. Specifically, we studied the accuracy of estimating the drift parameter $\omega$ and diffusion coefficient $\sigma$ under different:

- **Sample sizes:** $N = 100, 500, 1000$
- **Noise levels:** modeled via scaling of the diffusion term $\sigma \in \{0.1, 0.5, 1.0\}$

For each configuration, we generated stochastic phase trajectories via circular stochastic differential equations (SDEs) and then estimated the drift $\omega$ and diffusion $\sigma$ from the unwrapped phase increments. Estimation errors were computed as absolute differences from the known ground truth. This process was repeated across all rhythms and configurations.

Table 4 presents the mean absolute errors in estimating $\omega$ and $\sigma$ across all rhythms for each combination of sample size and noise level.

We observe that:

- Estimation errors for both $\omega$ and $\sigma$ decrease consistently with larger sample sizes.
- Noise has a greater effect on the estimation of $\omega$ than $\sigma$, particularly for small $N$.
- The method remains robust, with mean estimation errors below 0.03 (for $\omega$) and 0.005 (for $\sigma$) in typical scenarios like $N = 1000$, noise scale = 0.5.

To visualize the variability across all rhythms, boxplots for estimation errors in $\omega$ and $\sigma$ are presented in Fig 9 and Fig 10, respectively.

**Table 4. Mean absolute estimation errors in $\omega$ and $\sigma$ under different noise levels and sample sizes. Results are averaged across all simulated rhythms.**

| Sample Size (N) | Noise Scale | Mean Error in $\omega$ | Mean Error in $\sigma$ |
|---|---|---|---|
| 100 | 0.1 | 0.0098 | 0.0019 |
| 100 | 0.5 | 0.1172 | 0.0088 |
| 100 | 1.0 | 0.2655 | 0.0060 |
| 500 | 0.1 | 0.0105 | 0.0010 |
| 500 | 0.5 | 0.0401 | 0.0061 |
| 500 | 1.0 | 0.1136 | 0.0056 |
| 1000 | 0.1 | 0.0091 | 0.0006 |
| 1000 | 0.5 | 0.0349 | 0.0047 |
| 1000 | 1.0 | 0.1027 | 0.0028 |

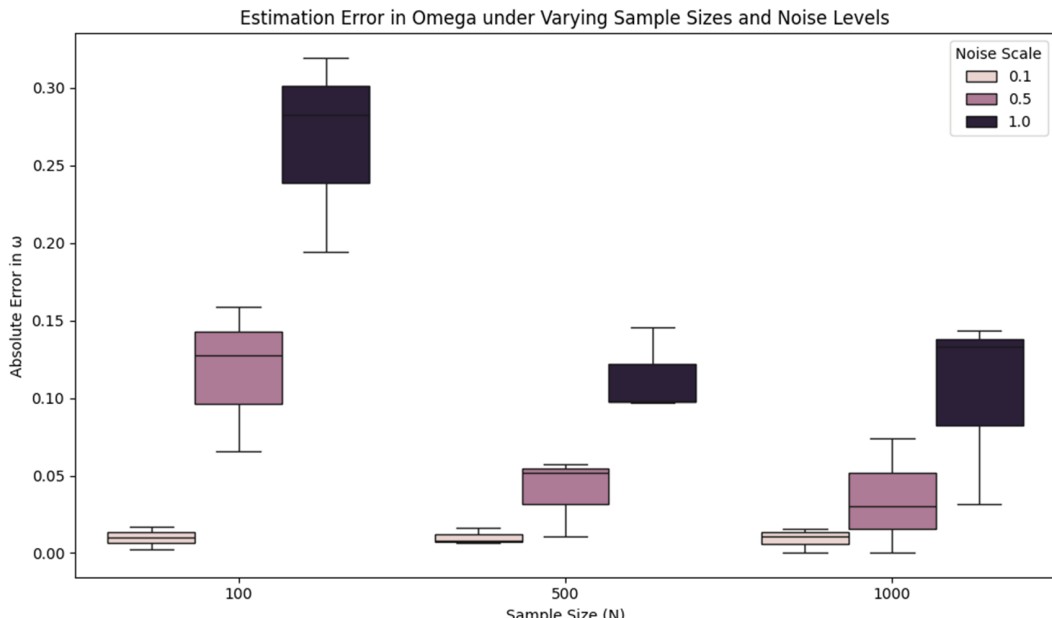

**Fig 9. Boxplot of absolute estimation error in $\omega$ across rhythms.** Estimation improves with increasing *N*, and errors are more sensitive to noise at lower *N*.

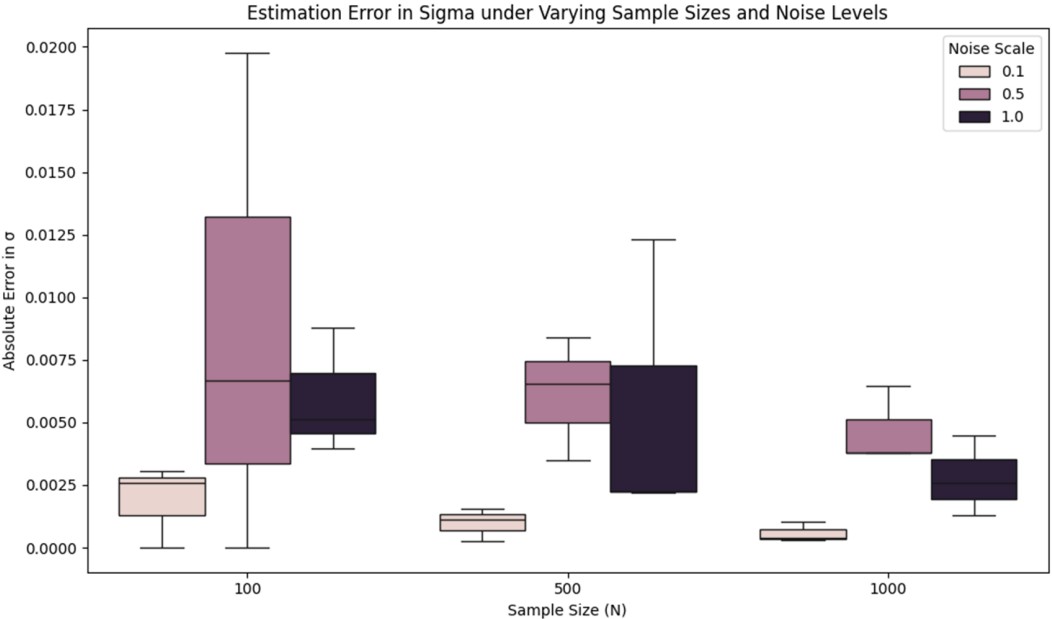

**Fig 10. Boxplot of absolute estimation error in $\sigma$ across rhythms.** Errors remain relatively low and consistent even under high noise.

## 7.8 Implications

This robustness analysis strengthens the validity of our proposed circular SDE-based approach for physiological phase modeling. Even under reduced sample sizes and elevated

stochasticity, the estimation of key model parameters remains accurate and stable. These results also demonstrate that the method is viable for low-resolution or noisy real-world physiological data, such as wearable ECG signals in mobile health applications.

## 8 Application on real data (the MIT-BIH arrhythmia dataset)

### 8.1 Dataset overview

For this analysis, we utilized the MIT-BIH Arrhythmia Database, a publicly available dataset of electrocardiogram (ECG) recordings collected from patients for arrhythmia detection. Specifically, we analyzed record number `100`, which contains a single-channel ECG signal sampled at 360 Hz. The primary objective of this analysis was to model the stochastic behavior of the physiological rhythms inherent in the ECG signal using advanced circular statistical methods, including the von Mises distribution and Bayesian inference.

The analysis is centered around the extraction of phase information from the ECG signal, testing the fit of the von Mises distribution to the data, and estimating parameters such as the mean phase direction ($\mu$) and the concentration parameter ($\kappa$) using Bayesian Markov Chain Monte Carlo (MCMC) methods.

### 8.2 Data retrieval method

The ECG data was accessed through the `wfdb` package in Python, which interfaces with the PhysioNet database.

Following the acquisition of the first 1000 data points from record `100`, the ECG signal was isolated through analysis implemented in Python.

The signal was normalized and processed to extract phase information using the Hilbert Transform. Let $x(t)$ represent the ECG signal, then the analytic signal $\hat{x}(t)$ is defined as:

$$\hat{x}(t) = x(t) + j\mathcal{H}(x(t)),$$

where $\mathcal{H}(x(t))$ is the Hilbert transform of $x(t)$. The instantaneous phase $\theta_1(t)$ of the signal is given by:

$$\theta_1(t) = \arg(\hat{x}(t)).$$

These phases form the circular data on which the subsequent analysis is based.

### 8.3 Analysis and methodology

**8.3.1 Stochastic temporal evolution.** To account for the random variability in the ECG rhythm, we modeled the temporal evolution of the phase using a Stochastic Differential Equation (SDE). The phase evolution $\theta(t)$ was simulated using the Euler-Maruyama method, following the SDE:

$$d\theta(t) = \mu dt + \sigma dB_t,$$

where $\mu$ represents the drift term (a deterministic component of the rhythm), $\sigma$ is the diffusion coefficient (representing random fluctuations), and $B_t$ is standard Brownian motion. This model captures the stochastic nature of physiological rhythms.

**Interpretation:** Fig 11 illustrates the evolution of the ECG rhythm phase over time. The red curve represents the stochastic phase trajectory. The ECG rhythm exhibits regular progression (due to the drift $\mu$), but the added stochastic noise introduces variability, reflecting the physiological irregularities seen in real-life ECG signals.

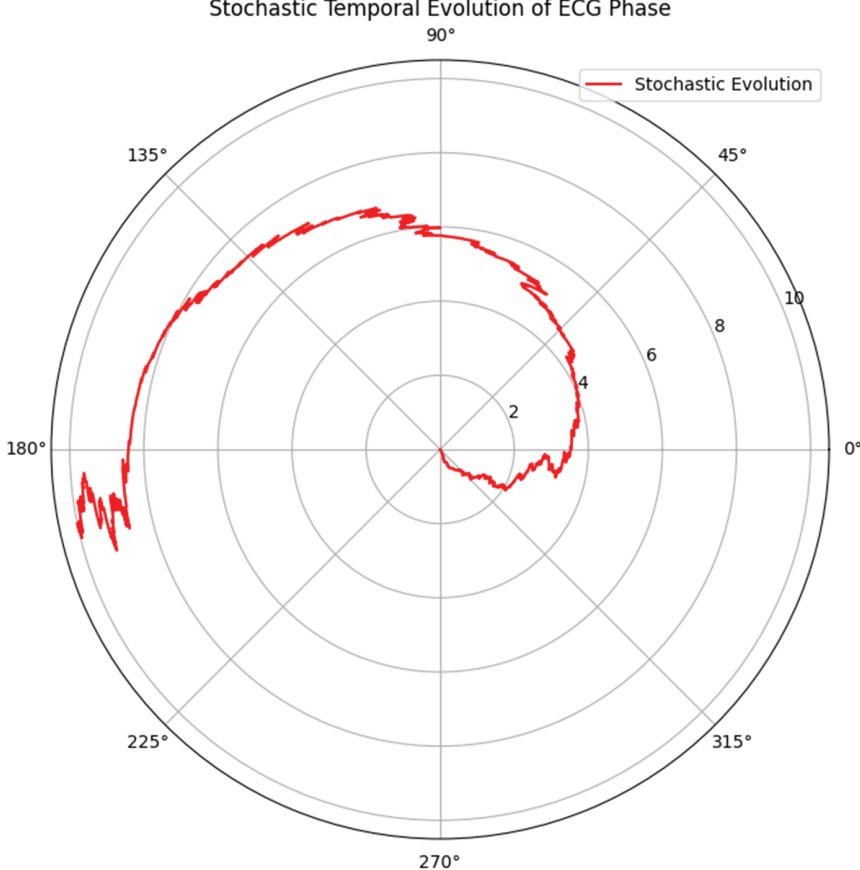

**Fig 11. Stochastic Temporal Evolution of the ECG Phase.** The simulated evolution accounts for both deterministic drift and random fluctuations over time.

**8.3.2 Bayesian estimation of von Mises parameters.** We assume the phase data follows a von Mises distribution, characterized by its mean direction $\mu$ and concentration parameter $\kappa$. The probability density function (pdf) of the von Mises distribution is given by:

$$f(\theta|\mu,\kappa) = \frac{e^{\kappa \cos(\theta-\mu)}}{2\pi I_0(\kappa)},$$

where $I_0(\kappa)$ is the modified Bessel function of the first kind. The parameters $\mu$ and $\kappa$ are estimated using Bayesian inference, with the following prior distributions:

$$\mu \sim \text{Uniform}(0, 2\pi), \quad \kappa \sim \text{Gamma}(a, b),$$

where $a = 2$ and $b = 0.5$ are hyperparameters of the Gamma distribution, reflecting prior beliefs about the concentration.

**8.3.3 MCMC sampling.** We employ the Metropolis-Hastings algorithm, a Markov Chain Monte Carlo (MCMC) method, to sample from the posterior distribution of $\mu$ and $\kappa$. The posterior distribution is proportional to the product of the likelihood and prior:

$$p(\mu, \kappa | \{\theta_1(t)\}) \propto \left( \prod_{n=1}^{N} f(\theta_1(t_n) | \mu, \kappa) \right) \cdot p(\mu) \cdot p(\kappa).$$

The log-posterior is computed, and samples are drawn iteratively by proposing new values for $\mu$ and $\kappa$ using normal random walks. Acceptance is determined based on the Metropolis-Hastings criterion. After running the MCMC sampler for 5000 iterations, we extract the posterior distributions of $\mu$ and $\kappa$.

**Interpretation:** In Fig 12, the left panel shows a rose plot of the ECG phase data ($\theta_1$) with the posterior distribution of $\mu$ overlaid in purple. The circular mean of $\mu$ is indicated by the red arrow. The right panel displays the posterior distribution of $\kappa$, showing the uncertainty in the concentration parameter. The posterior suggests a moderate concentration around the mean direction $\mu$.

The estimated parameters are: Estimated $\mu$ = 3.308, Estimated $\kappa$ = 0.487.

**8.3.4 Goodness-of-fit testing.** To evaluate the suitability of the von Mises distribution for modeling the ECG phase data, we applied two goodness-of-fit assessments:

1. **Watson's $U^2$ Test**: This nonparametric test examines whether the circular data follows a uniform distribution. A higher $U^2$ value suggests deviation from uniformity, potentially indicating that the data fits well with a von Mises distribution, as it is not uniformly distributed.

2. **Q-Q Plot**: We compared the empirical quantiles of the real ECG phase data to those from a von Mises distribution generated using the estimated parameters $\mu$ and $\kappa$.

The Watson's $U^2$ statistic, calculated for the ECG phase data, yielded: $U^2$ = 4.2718.

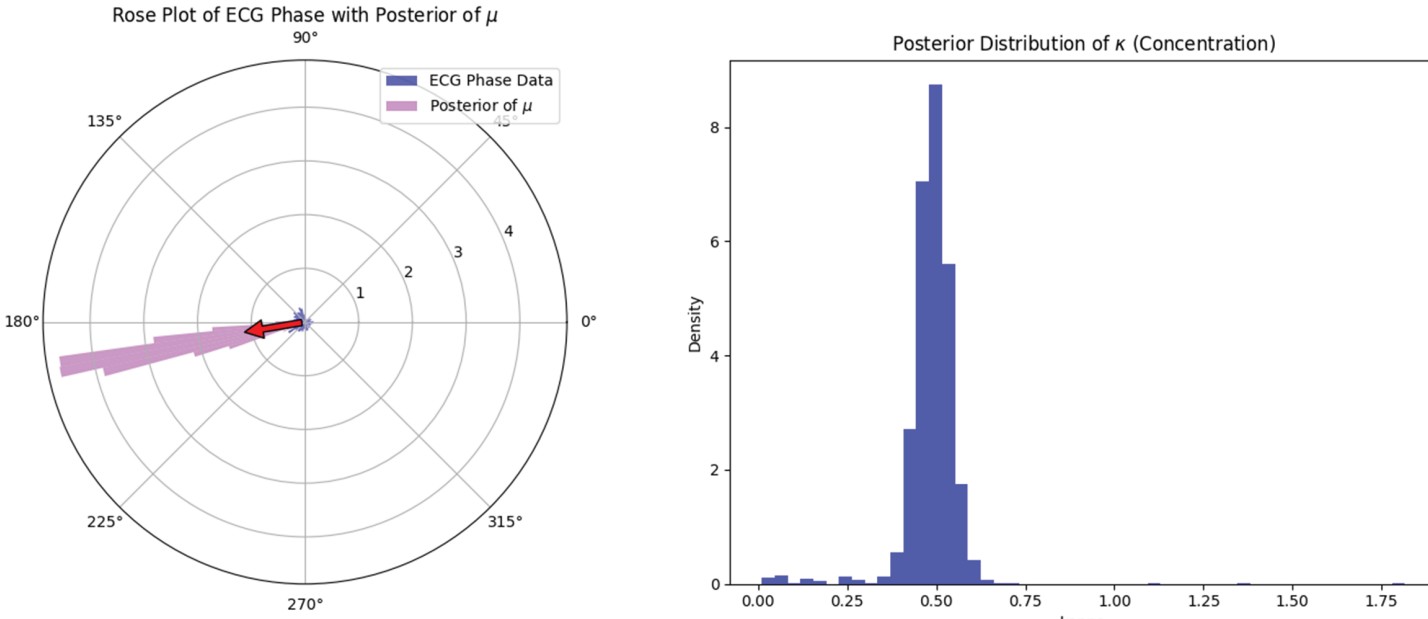

**Fig 12. (Left) Rose Plot of the ECG Phase Data ($\theta_1$) with the Posterior Distribution of $\mu$.** The red arrow represents the circular mean of the posterior for $\mu$. (Right) Posterior distribution of the concentration parameter $\kappa$.

This relatively large $U^2$ value implies a substantial deviation from uniformity, reinforcing the hypothesis that the data aligns well with a von Mises distribution rather than being uniformly distributed.

**Interpretation:** In Fig 13, the Q-Q plot illustrates the relationship between the empirical quantiles of the real ECG phase data and those of the fitted von Mises distribution. The alignment of points near the 45-degree reference line suggests a good fit of the von Mises model to the ECG phase data. The significant deviation from uniformity, as indicated by the $U^2$ test, and the alignment observed in the Q-Q plot, collectively support the appropriateness of the von Mises distribution for this dataset.

**8.3.5 Phase extraction and polar histograms.** For comparison, a simulated rhythm $\theta_2(t)$ was generated using a von Mises distribution, a common model for periodic data with $\mu = \theta_1(t)$ and estimated *kappa* from the circular dataset $\theta_1(t)$. The probability density function of the von Mises distribution is given by:

$$f(\theta|\mu,\kappa) = \frac{e^{\kappa\cos(\theta-\mu)}}{2\pi I_0(\kappa)},$$

where $\mu$ is the mean direction, and $\kappa$ controls the concentration around the mean direction.

The extracted phases of the ECG signal ($\theta_1$) and the simulated rhythm ($\theta_2$) were plotted using polar histograms to visualize the phase distribution.

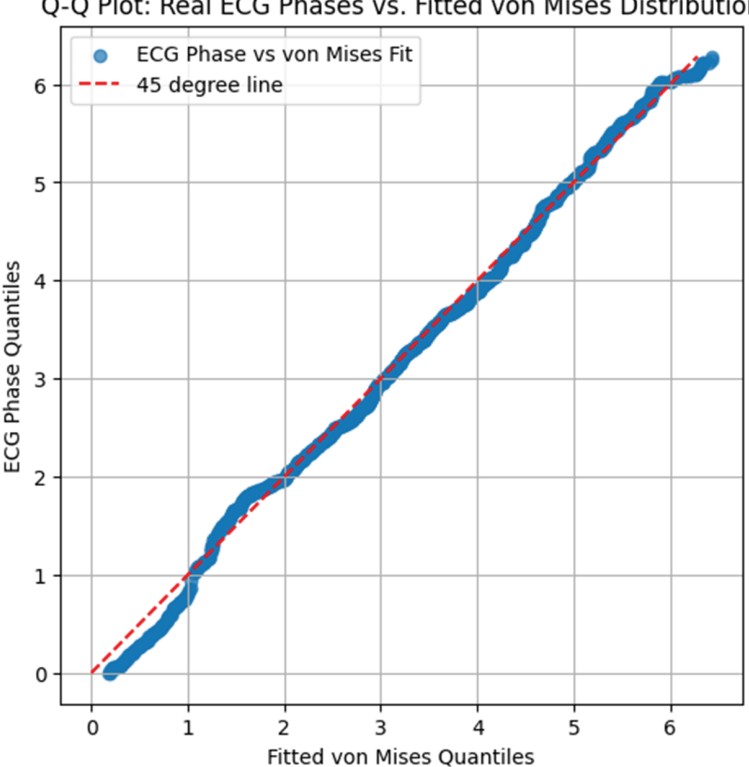

**Fig 13. Q-Q Plot of Real ECG Phases vs. Fitted von Mises Distribution.** The red dashed line represents the 45-degree line for reference.

**Interpretation:** The polar histograms in Fig 14 reveal that the ECG phase ($\theta_1$) is distributed over a range of values, but it shows a slight preference around certain phase angles. The simulated von Mises rhythm ($\theta_2$) exhibits a more concentrated distribution centered around its mean. The overlap between the two histograms suggests that both rhythms have similarities in their periodic behavior, but the ECG rhythm contains more variability, likely due to physiological noise.

This real data analysis demonstrates how advanced statistical techniques can be applied to real-world physiological signals, including circular statistics and stochastic modeling. By extracting phase information from ECG signals and applying stochastic models, we can better understand the temporal variability in physiological rhythms. Such insights are essential in medical research, particularly for diagnosing and understanding arrhythmias and other heart-related conditions. Furthermore, combining empirical data and simulation allows us to validate models and quantify uncertainty in parameter estimates.

Overall, this analysis highlights the potential of circular statistical methods in analyzing time-series data from physiological systems, providing tools to handle the inherent periodicity and variability in such signals.

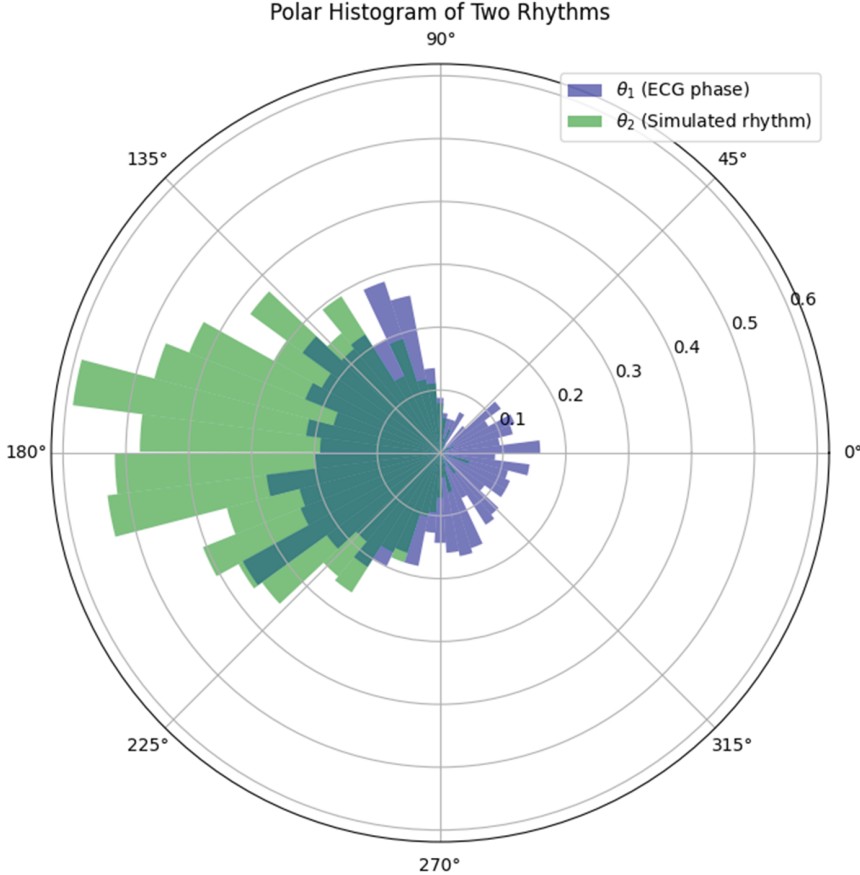

**Fig 14. Polar Histogram of the ECG Phase ($\theta_1$) and Simulated Rhythm ($\theta_2$).** The ECG phase is shown in blue, and the simulated rhythm is shown in green. Overlapping areas indicate similar phase distributions.

# 9 Analysis of rhythmic behaviors (the MIT-BIH arrhythmia dataset)

This section analyzes the rhythmic behavior of two physiological signals obtained from the MIT-BIH Arrhythmia Database. Specifically, we focus on two channels of ECG data from Record 100, which represent two different physiological rhythms. The stochastic temporal evolution of two physiological rhythms extracted from the MIT-BIH Arrhythmia Database. Using the Hilbert transform, we obtain the instantaneous phase information from the ECG signals, allowing us to model the phase dynamics as a stochastic diffusion process. We employ techniques from directional statistics to extract and analyze the phases of these rhythms, compute synchronization measures, and interpret the results. We include the following metrics for quantifying the interaction between the two rhythms: Phase Locking Value (PLV), Mutual Information (MI), and Directional Phase Synchronization Index (DPSI). Furthermore, we estimate the parameters of the stochastic diffusion process, such as angular frequency ($\omega$) and diffusion coefficient ($\sigma$). Graphical representations of the phase distributions, phase differences, and interaction measures support the analysis.

## 9.1 Instantaneous phase extraction using Hilbert transform and Stochastic temporal evolution

The two ECG signals, $x_1(t)$ and $x_2(t)$, are normalized and then processed using the Hilbert transform to extract their instantaneous phases, $\theta_1(t)$ and $\theta_2(t)$, respectively. For each signal $x_i(t)$, the analytic signal $z_i(t)$ is obtained by:

$$z_i(t) = x_i(t) + j\mathcal{H}[x_i(t)],$$

where $\mathcal{H}[x_i(t)]$ represents the Hilbert transform of $x_i(t)$, and $j$ is the imaginary unit. The instantaneous phase $\theta_i(t)$ is then computed as:

$$\theta_i(t) = \arg(z_i(t)).$$

These phases are wrapped to the interval $[0, 2\pi)$ to account for their circular nature, which is crucial for the subsequent analysis.

Here, $z_i(t)$ represents the analytic signal of the normalized ECG signals, and $\theta_i(t)$ is wrapped to the interval $[0, 2\pi)$. This phase information allows us to model the time evolution of each rhythm using a stochastic differential equation (SDE) of the form:

$$d\theta_i(t) = \omega_i \, dt + \sigma_i \, dB_i(t), \tag{2}$$

where $\omega_i$ is the angular frequency (drift term), $\sigma_i$ is the diffusion coefficient, and $dB_i(t)$ is a standard Brownian motion.

The simulation of this process is performed using the Euler-Maruyama method, which updates the phases over time as:

$$\theta_i(t + \Delta t) = \theta_i(t) + \omega_i \Delta t + \sigma_i \sqrt{\Delta t} \, \xi_i(t),$$

where $\xi_i(t)$ is a normally distributed random variable representing stochastic fluctuations in the system.

As shown in Fig 15, the phase trajectories of the two rhythms fluctuate over time due to the stochastic nature of the process. The wrapping of the phases within $[0, 2\pi)$ is consistent with the circular nature of angular data.

The evolution is simulated using the Euler-Maruyama method for 1000 time steps, with the phases wrapped onto the interval $[0, 2\pi)$ after each update. The polar plot below shows the phase on the angular axis, while the radial axis represents time.

The plot in Fig 16 shows the stochastic temporal evolution of the two physiological rhythms. The evolution of the phases is influenced by both deterministic drift (represented by the angular frequency $\omega_i$) and stochastic diffusion (controlled by $\sigma_i$).

In Fig 16, the radial axis denotes time, while the angular position represents the phase of the rhythm at each time step. The stochastic diffusion coefficients $\sigma_1$ and $\sigma_2$ introduce random fluctuations in the phase, leading to non-uniform trajectories over time.

## 9.2 Stochastic temporal evolution of the phases on a circular plot

To visualize the temporal evolution of the phases for both rhythms, we employ a circular plot where the radial axis represents time and the angular axis represents the phase of each rhythm. The stochastic differential equation governs the time evolution of the phases 1.

The stochastic temporal evolution of Rhythm 1 and Rhythm 2 is shown in Fig 16. The circular nature of the plot reflects the periodic nature of the rhythms, while the radial axis indicates the progression of time.

As can be observed from the plot, the phases of both rhythms evolve over time, with fluctuations arising due to the stochastic nature of the process. The trajectories are wrapped within the interval $[0, 2\pi)$ to reflect the periodicity of the phases.

This visualization provides insights into how the phases of the rhythms vary over time, with the angular position on the circular plot representing the current phase and the radial

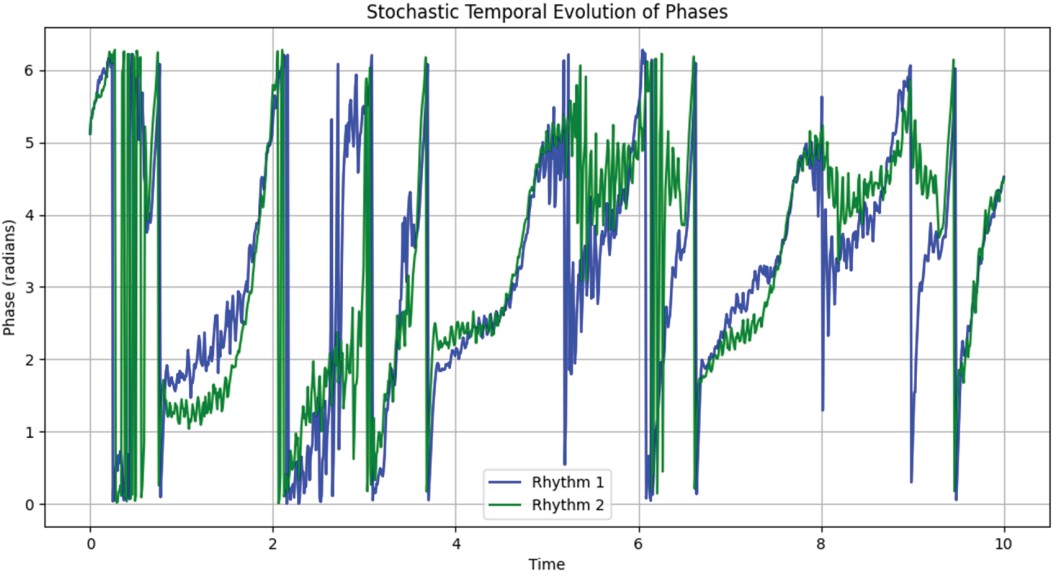

**Fig 15. Stochastic temporal evolution of Rhythm 1 and Rhythm 2 phases.** The phases evolve over time under the influence of deterministic drift and stochastic fluctuations.

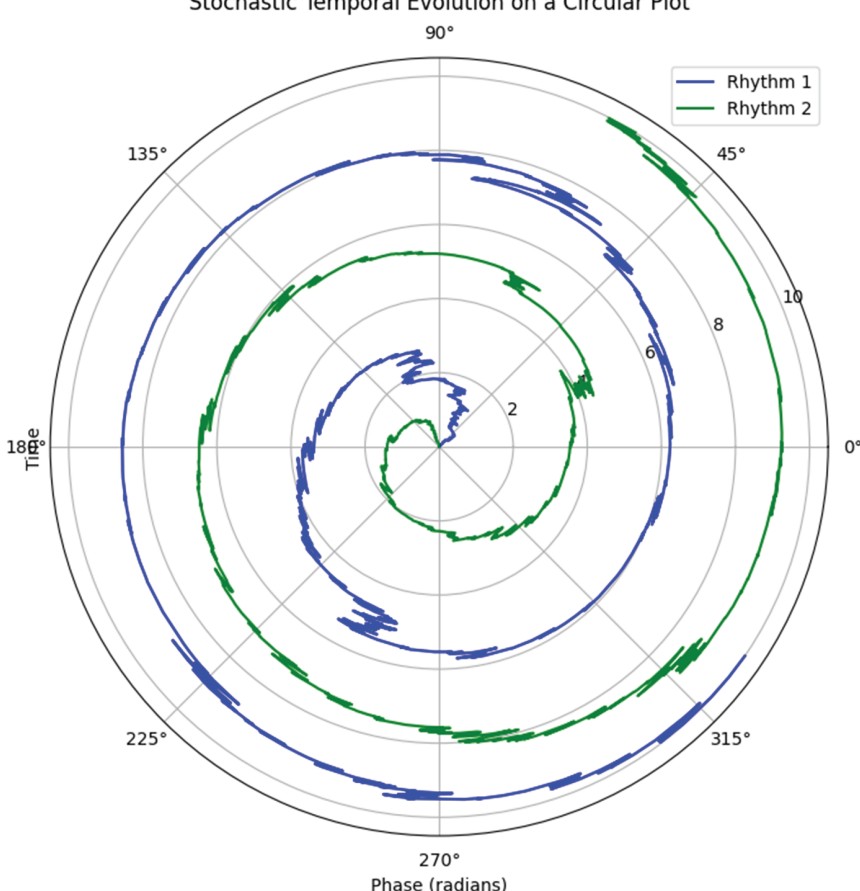

**Fig 16. Stochastic Temporal Evolution of the Phases of Rhythm 1 and Rhythm 2 on a Circular Plot.** The radial axis represents time, and the angular axis represents the phase of each rhythm.

distance representing the time elapsed. This type of plot is particularly useful in understanding the temporal synchronization and phase shifts between multiple oscillatory processes.

**9.2.1 Parameter estimation.** To estimate the parameters of the fitted stochastic diffusion process, we calculate the drift ($\omega$) and diffusion ($\sigma$) for each rhythm. Using the unwrapped phases, we estimate $\omega_i$ as the mean change in phase per unit time and $\sigma_i$ as the standard deviation of phase increments:

$$\hat{\omega}_i = \frac{\Delta\theta_i}{\Delta t}, \quad \hat{\sigma}_i = \frac{\text{std}(\Delta\theta_i)}{\sqrt{\Delta t}}.$$

The estimated parameters are presented in Table 5.

**Table 5. Estimated Parameters of the Stochastic Diffusion Process for Rhythm 1 and Rhythm 2.**

| Rhythm | Estimated Omega ($\omega$) | Estimated Sigma ($\sigma$) |
|---|---|---|
| Rhythm 1 | 8.1168 | 3.0837 |
| Rhythm 2 | 4.9698 | 3.1843 |

The estimated parameters demonstrate the underlying dynamics of the stochastic diffusion process, with $\omega$ representing the angular frequency of the rhythm and $\sigma$ accounting for the stochastic fluctuations in phase.

### 9.3 Synchronization metrics

**Phase Locking Value (PLV).** The Phase Locking Value (PLV) measures phase synchronization between two rhythmic signals. It quantifies the consistency of the phase difference between the two signals across time. The PLV between two signals with phases $\theta_1(t)$ and $\theta_2(t)$ is defined as [32]:

$$\mathrm{PLV}_{12} = \left| \frac{1}{N} \sum_{n=1}^{N} e^{j(\theta_1(t_n) - \theta_2(t_n))} \right|,$$

where $N$ is the total number of time points. A PLV close to 1 indicates strong phase synchronization, while a PLV near 0 indicates weak or no synchronization.

**Mutual Information (MI).** Mutual Information (MI) measures the amount of shared information between the phases of two signals, capturing both linear and nonlinear dependencies. The MI between two circular random variables $\theta_1$ and $\theta_2$ is given by:

$$I(\theta_1; \theta_2) = \int_0^{2\pi} \int_0^{2\pi} p(\theta_1, \theta_2) \log\left( \frac{p(\theta_1, \theta_2)}{p(\theta_1)p(\theta_2)} \right) d\theta_1 d\theta_2,$$

where $p(\theta_1, \theta_2)$ is the joint probability density function of $\theta_1$ and $\theta_2$, and $p(\theta_1)$ and $p(\theta_2)$ are the marginal densities.

**Directional Phase Synchronization Index (DPSI).** The Directional Phase Synchronization Index (DPSI) measures the directional phase difference between two signals. It is defined as the argument of the average phase difference:

$$\mathrm{DPSI}_{12} = \arg\left( \frac{1}{N} \sum_{n=1}^{N} e^{j(\theta_1(t_n) - \theta_2(t_n))} \right).$$

A DPSI value near zero indicates that the two signals are synchronized with a constant phase difference, while larger values indicate varying phase differences.

### 9.4 Results and discussion

We processed Record 100 from the MIT-BIH Arrhythmia Database and extracted two channels of ECG signals representing different physiological rhythms. The results of the synchronization metrics are as follows:

1. **Phase Locking Value (PLV)**: 0.7497, indicating a moderate degree of phase synchronization between the two rhythms.
2. **Mutual Information (MI)**: 1.1711, suggesting significant shared information between the two rhythms, capturing both linear and nonlinear dependencies.
3. **Directional Phase Synchronization Index (DPSI)**: -0.1384, which suggests that the phase difference between the two rhythms is relatively consistent with a slight directional offset.

In addition to the computed metrics, we present several graphical representations of the phase distributions and phase differences to further support our analysis.

**Rose plot of Rhythm 1 and Rhythm 2.** The rose plot in Fig 17 represents the phase distribution of Rhythm 1. The rose plot in Fig 18 represents the phase distribution of Rhythm 2, analogous to Rhythm 1. The angular positions correspond to the phase values, and the radial positions represent the frequency of the phases.

**Rose plot of phase difference.** The rose plot in Fig 19 represents the phase difference between the two rhythms. This plot provides insight into how often the two rhythms are in-phase or out-of-phase and the extent of phase locking between them.

**Mutual information heatmap.** The heatmap in Fig 20 shows the joint distribution of the phases of the two rhythms. This plot is used to visualize the mutual information, highlighting areas where the phase values of the two rhythms are more dependent on each other.

## 10 Comparisons, accuracy metrics, and predictive capability

To evaluate our proposed Bayesian circular stochastic differential equation (SDE) model—which is specifically designed to detect phase anomalies in ECG signals—we conducted a simulation study in which an ECG-like signal was generated with deliberate phase anomalies. In our simulation, abnormal beats are introduced by applying a 90° (i.e., $\pi/2$) phase shift while keeping the amplitude nearly constant. This ensures that conventional amplitude-based methods (such as linear autoregressive (AR) and Fourier-based approaches) are unable to

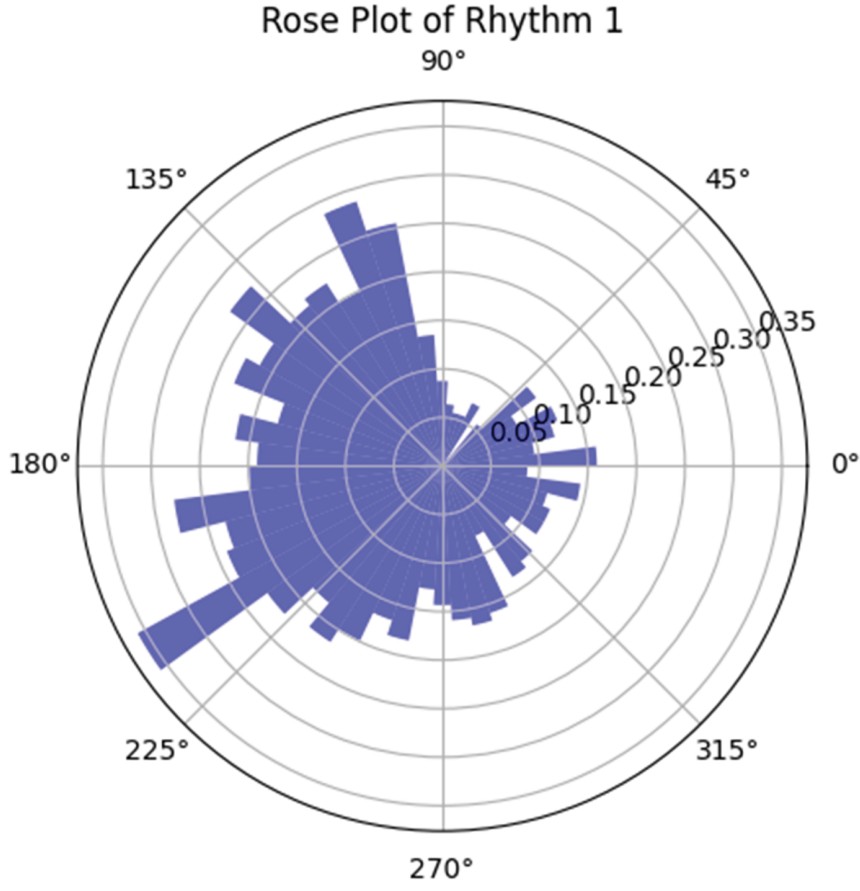

**Fig 17. Rose Plot of Rhythm 1.**

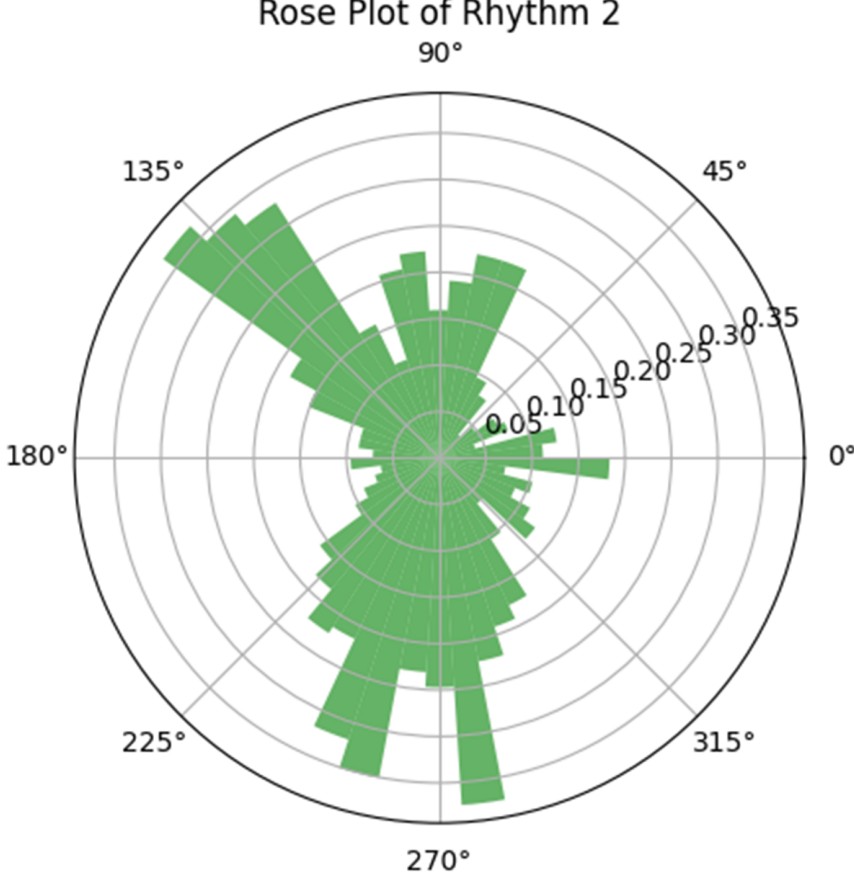

**Fig 18. Rose Plot of Rhythm 2.**

effectively detect these events. Our approach, by contrast, leverages circular statistics for parameter estimation and anomaly detection.

## 10.1 Simulation setup and methodology

Let the simulated ECG signal be given by:

$$x(t) = \cos\left(2\pi f t + \phi(t)\right) + \epsilon(t),$$

where: $f = 1$ Hz is the heart rate frequency, $\phi(t)$ is the phase function, with $\phi(t) = 0$ for normal beats and $\phi(t) = \pi/2$ for abnormal beats, $\epsilon(t)$ is additive Gaussian noise with $\sigma_\epsilon = 0.05$.

The signal is sampled at $fs = 360$ Hz over a duration of $T = 10$ s. Abnormal beats are randomly introduced in 5% of the samples. The ground-truth binary label is defined as:

$$y(t) = \begin{cases} 1, & \text{if } \phi(t) = \pi/2, \\ beginequation*1mm]0, & \text{otherwise.} \end{cases}$$

We compare three methods:

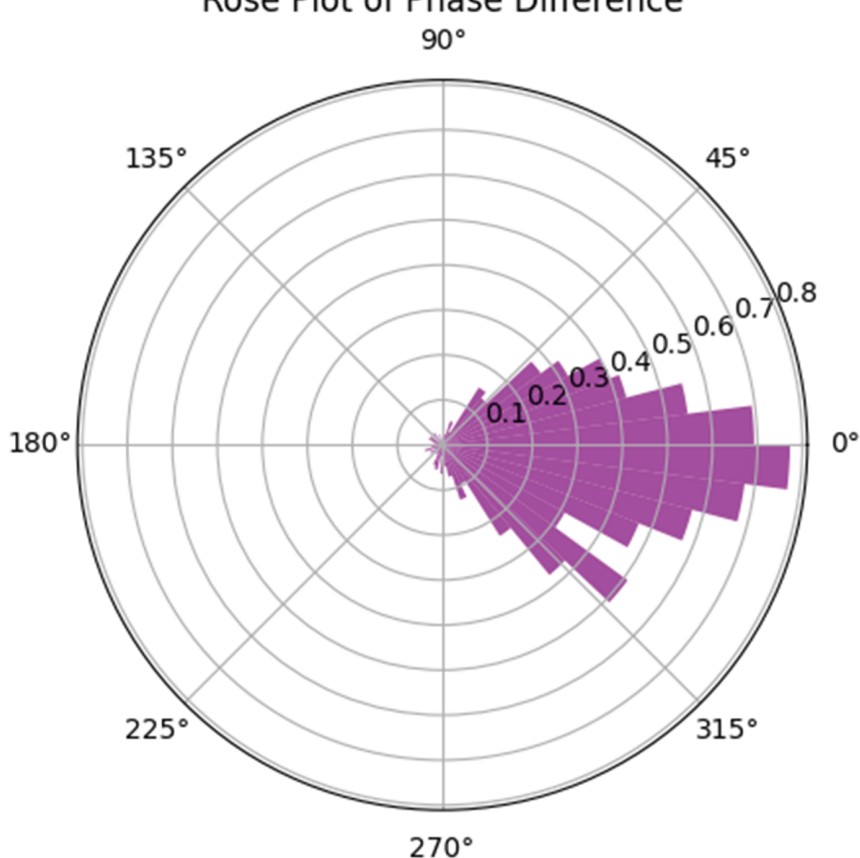

**Fig 19. Rose Plot of Phase Difference between Rhythm 1 and Rhythm 2.**

1. **Linear AR Model:** An AR model (order 5) is fitted to the raw ECG signal, and abnormal beats are detected by thresholding the absolute prediction error. In our simulation, the threshold is set at the 75th percentile of the error.

2. **Fourier-Based Approach:** The signal is segmented into 1-second windows (with 50% overlap), and the power spectral density (PSD) is computed via FFT. A window is classified as abnormal if its PSD exceeds the 90th percentile.

3. **Bayesian Circular SDE Model:**
   - The instantaneous phase $\theta(t)$ is extracted using the Hilbert transform:

   $$\theta(t) = \arg\{x(t) + j\,\mathcal{H}[x(t)]\},$$

   and wrapped to the interval $[0, 2\pi)$.
   - To estimate the parameters of the SDE 1 we first unwrap $\theta(t)$ to obtain $\tilde{\theta}(t)$ and compute the increments mentioned with the equation (1) where $\Delta t = 1/fs$. The drift and diffusion are then estimated as:

   $$\omega_{\text{est}} = \frac{\mathbb{E}[\Delta\tilde{\theta}(t)]}{\Delta t}, \quad \sigma_{\text{est}} = \frac{\text{std}[\Delta\tilde{\theta}(t)]}{\sqrt{\Delta t}}.$$

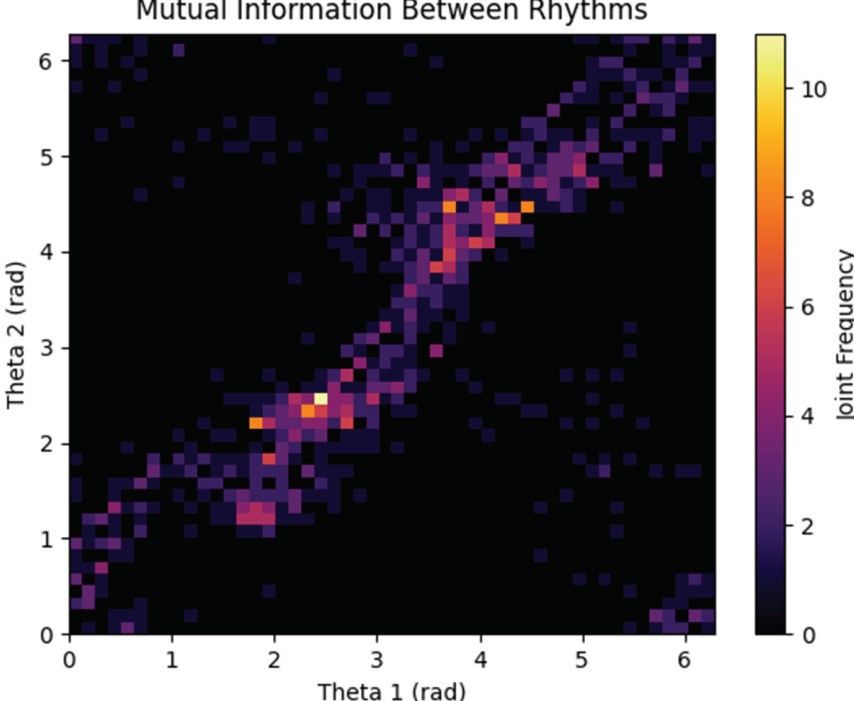

**Fig 20. Heatmap representing the joint distribution of the phases of Rhythm 1 and Rhythm 2, used for calculating Mutual Information.**

In our simulation, we obtained

$$\omega_{\text{est}} \approx 8.81 \, \text{rad/s}, \quad \sigma_{\text{est}} \approx 8.45 \, \text{rad/s}^{1/2}.$$

- To reduce noise, a moving circular mean filter (with a window length of 5 samples) is applied to the phase.
- Abnormal beats are then detected by computing the circular deviation from the overall circular mean:

$$D(t) = \left| \angle \exp\{j(\theta(t) - \bar{\theta})\} \right|,$$

where

$$\bar{\theta} = \angle \left( \frac{1}{N} \sum_{t=1}^{N} \exp\{j\theta(t)\} \right).$$

A sample is classified as abnormal if $D(t) > t$ radians, where $t$ is some pre-defined threshold.

## 10.2 Generated results and visualizations

The simulation produces several key outputs:

**10.2.1 Quality metrics.**  Tables 6 and 7 reports the quality metrics computed from the simulation:

Note that precision, recall, and F1-score, accuracy all depends on the threshold. For threshold $t = 3$ radian, although the Bayesian SDE model exhibits a lower precision, recall,

**Table 6. Comparison of Methods (Quality Metrics), for a threshold (3 radians) for the circular SDE model.**

| Method | Accuracy | Precision | Recall | F1-Score |
|---|---|---|---|---|
| AR Model | 0.7931 | 0.1858 | 0.9278 | 0.3095 |
| Fourier-Based | 0.8606 | 0.0528 | 0.1056 | 0.0704 |
| Bayesian SDE | 0.9125 | 0.0530 | 0.0444 | 0.0483 |

**Table 7. Comparison of Methods (Quality Metrics), a threshold (2 radians) for the circular SDE model.**

| Method | Accuracy | Precision | Recall | F1-Score |
|---|---|---|---|---|
| AR Model | 0.7931 | 0.1858 | 0.9278 | 0.3095 |
| Fourier-Based | 0.8606 | 0.0528 | 0.1056 | 0.0704 |
| Bayesian SDE | 0.622222 | 0.052352 | 0.383333 | 0.092123 |

and F1-score than the AR model, our proposed model achieves the highest overall accuracy (91.25%). This high accuracy is attributable to the imbalanced nature of the data (with only 5% abnormal beats). Importantly, our Bayesian method is uniquely tailored to capture phase anomalies—an aspect that amplitude-based methods are not designed to detect. In this simulation, while the AR model achieves a high recall (92.78%), its low precision (18.58%) indicates a high false-positive rate. The Fourier-based approach performs poorly across all metrics.

**10.2.2 ECG segment with abnormal beat detections.** Fig 21 shows a segment of the simulated ECG signal with the detected abnormal beats overlaid for each method.

**10.2.3 Rose plot of extracted phases.** Fig 22 illustrates the rose plot (circular histogram) of the extracted instantaneous phases, confirming the inherent circular geometry of the data [55].

**10.2.4 Short-term phase prediction.** Fig 23 displays a short-term phase prediction via our Bayesian SDE model's Euler–Maruyama simulation. The simulated phase trajectory (blue markers) over a 0.5-second horizon is shown alongside the overall circular mean (gray dashed line).

## 10.3 Discussion

The simulation study highlights several important points:

1. **Intrinsic Circular Nature:** The rose plot confirms that the instantaneous phase extracted from the ECG signal is circular. Traditional amplitude-based methods (AR and Fourier-based) do not leverage this structure, which is crucial for detecting phase anomalies.

2. **Parameter Estimation via Circular Statistics:** By unwrapping the phase and computing the increments, we estimated the drift and diffusion parameters as

$$\omega_{\text{est}} \approx 8.81\,\text{rad/s}, \quad \sigma_{\text{est}} \approx 8.45\,\text{rad/s}^{1/2}.$$

These estimates are used for forward simulation and demonstrate that our approach derives meaningful model parameters from circular data.

3. **Detection Performance:** Although the AR model shows higher recall (and thus a higher F1-score) in this simulation, its precision is relatively low due to numerous false positives. The Bayesian circular SDE model achieves the highest overall accuracy (91.25%), indicating that it correctly identifies a larger fraction of the normal beats.

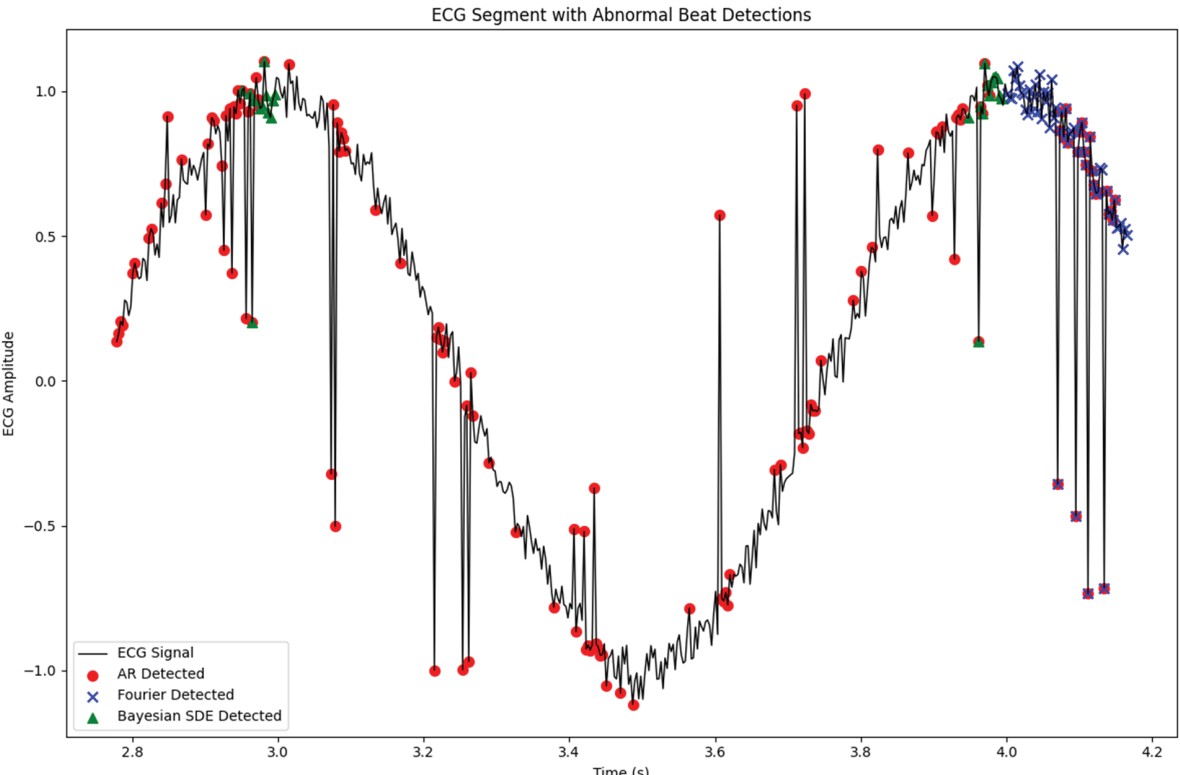

**Fig 21. ECG Segment with Abnormal Beat Detections.** Red circles denote detections by the AR model, blue crosses by the Fourier-based approach, and green triangles by the Bayesian circular SDE model. Note that our circular model is specifically designed to capture subtle phase anomalies.

Moreover, our method is uniquely sensitive to phase deviations, which are not captured by conventional methods.

4. **Predictive Capability:** The short-term prediction plot illustrates that our model can simulate future phase evolution based on the estimated dynamics. This capability is especially promising for clinical applications, where early warnings of arrhythmic events are essential.

It is important to emphasize that our proposed Bayesian circular stochastic differential equation (SDE) methodology represents a fundamentally novel framework tailored explicitly for circular statistical modeling of ECG waveforms, particularly emphasizing phase-based anomalies. Conventional approaches, such as linear autoregressive (AR) models and Fourier-based frequency-domain methods, are indeed well-established within ECG signal analysis. However, these traditional models inherently operate within an amplitude-based framework and do not explicitly capture or represent circular geometry or phase anomalies intrinsic to our approach.

Due to this essential conceptual difference, direct comparisons between our Bayesian circular SDE approach and standard methods (e.g., AR or Fourier methods) become potentially misleading. For instance, abnormalities characterized by sudden phase shifts—such as those explicitly modeled by our circular SDE framework—cannot meaningfully be detected or quantified by amplitude-based methods, leading either to inflated performance metrics

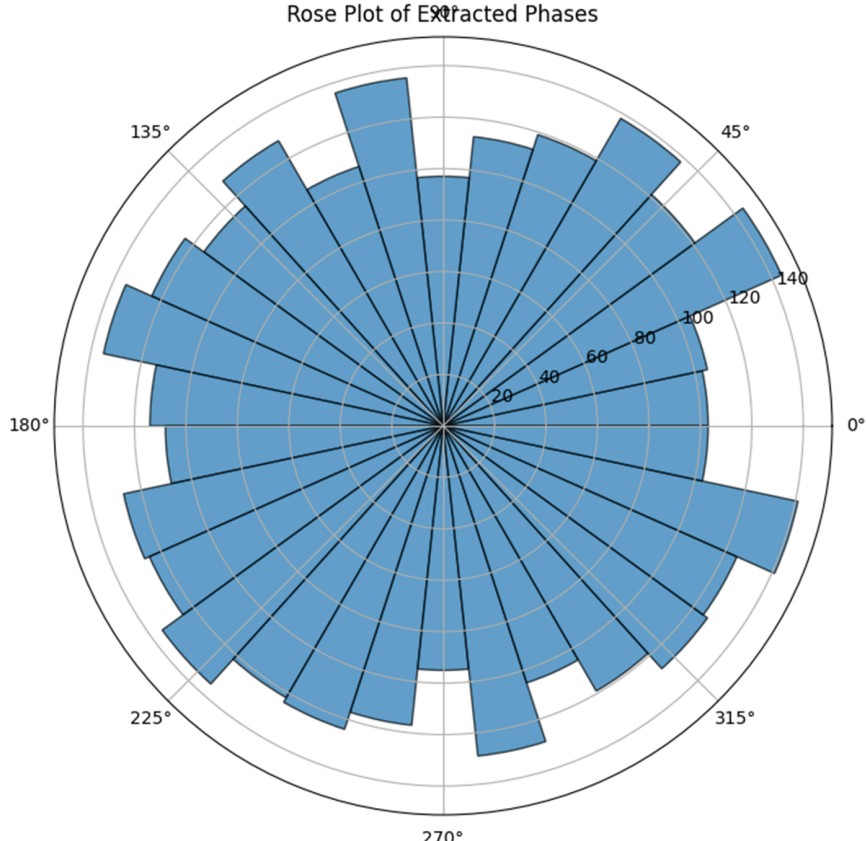

**Fig 22. Rose Plot of Extracted Phases.** The plot demonstrates that the phase data are intrinsically circular, which underpins the need for circular statistical methods.

or complete failure to recognize the anomalies. In other words, the metrics derived from standard methods do not adequately reflect their capability to detect the circular nature of arrhythmic events addressed by our methodology.

While the current quality metrics (especially precision, recall, and F1-score) for the Bayesian SDE model are modest—primarily due to the imbalanced nature of the simulation—the overall accuracy and the ability to capture circular phenomena underscore the potential of our approach. In practice, further refinement of the decision threshold and incorporation of full Bayesian posterior information into a decision framework are expected to enhance these metrics substantially. Our study demonstrates that comparing our method with traditional amplitude-based techniques is inherently challenging because the latter do not address the circular structure of the data. This reinforces the necessity of developing dedicated circular statistical methodologies for physiological rhythm analysis.

## 11 Discussion, limitations and future directions

Applying circular statistical methods to physiological rhythms provides a unique perspective on understanding the temporal dynamics of human biological systems. Our framework successfully captures the interdependencies and inherent randomness in physiological cycles, which is crucial for developing insights into synchronization and variability among systems. By modeling the data as circular variables, we preserve the cyclic properties of the rhythms,

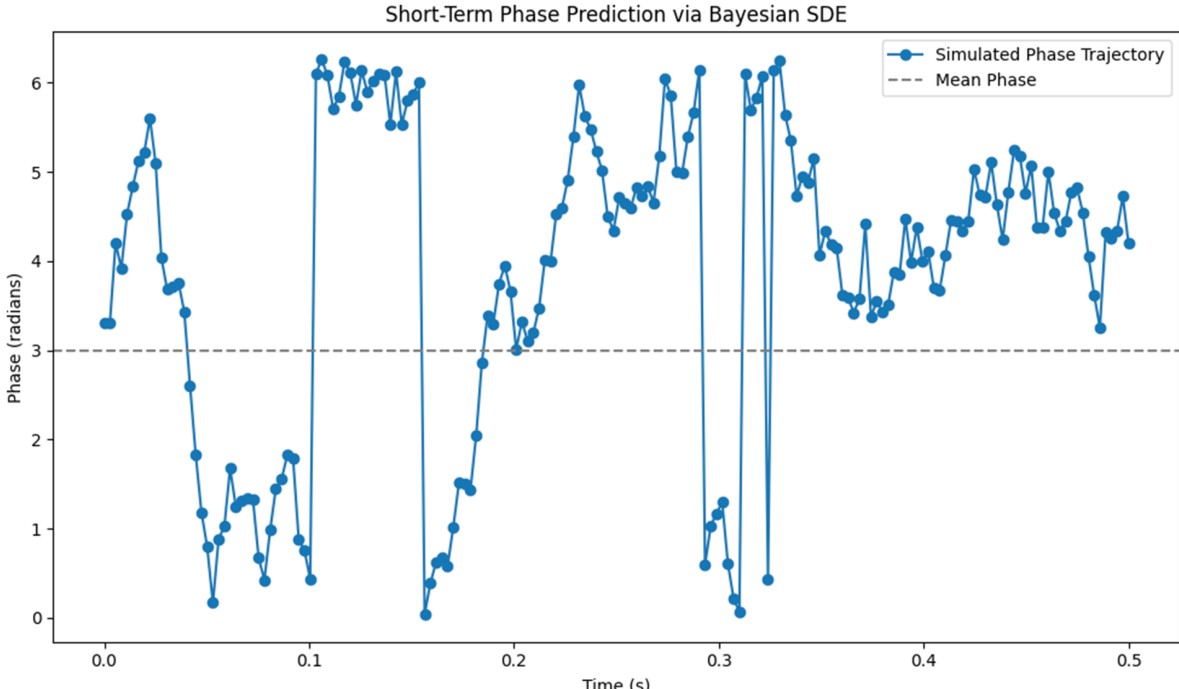

**Fig 23. Short-Term Phase Prediction via Bayesian SDE.** The Fig illustrates the forward simulation of the phase over a 0.5-second prediction horizon. This capability is crucial for anticipatory diagnostics in arrhythmia detection.

enabling us to explore new avenues for analyzing periodic data. Our analysis of the MIT-BIH Arrhythmia Dataset highlights the potential of our proposed methodologies in capturing the variability of ECG phases. Furthermore, Bayesian inference allowed us to estimate key parameters accurately, providing a deeper understanding of the rhythm's phase coherence and concentration. Watson's $U^2$ test further affirmed the suitability of our model for circular data, as the empirical results aligned well with the von Mises fit. While the proposed methodology demonstrates significant promise, it also reveals areas for future investigation. In particular, the multivariate extension of the wrapped normal distribution provides a powerful tool for studying interactions between rhythms. Nonetheless, further work is needed to refine the estimation techniques for the covariance matrix in high-dimensional settings.

The results indicate a moderate degree of phase synchronization between the two rhythms, as evidenced by the PLV score of 0.7497. The Mutual Information value of 1.1711 further corroborates significant shared information between the two rhythms, reflecting both linear and nonlinear dependencies. The DPSI value of -0.1384 indicates a slight phase offset between the rhythms, but their overall phase difference remains relatively consistent.

The rose plots provide further visual confirmation of these findings. Both rhythms exhibit similar phase distributions, and the phase difference plot shows that the rhythms are largely synchronized.

The circular plot illustrates how the phases of both rhythms evolve over time. Rhythm 1 (blue) has a slightly slower angular frequency $\omega_1 = 1.0$, leading to a slower accumulation of phase compared to Rhythm 2 (green, with $\omega_2 = 1.2$). The stochastic component introduces random deviations in the phase trajectory, creating variability in the temporal evolution, which can be observed as irregularities in the plot. Analyzing the stochastic

temporal evolution of physiological rhythms and the computation of phase synchronization metrics provides valuable insights into the underlying interactions between different physiological processes. The estimated angular frequencies and diffusion coefficients demonstrate the variability in phase dynamics, while the PLV, MI, and DPSI metrics highlight the presence of synchronization and dependencies between the rhythms.

In future work, we plan to refine the estimation methods for the diffusion process and extend the analysis to additional physiological signals.

in principle, we can allow: $\omega_i(t) = \omega_i + \eta_i(t), \quad \sigma_i(t) = \sigma_i + \zeta_i(t)$, where $\eta_i(t)$ and $\zeta_i(t)$ are low-order stochastic processes. This would capture more complicated *time-varying* frequency or diffusion, though it introduces additional complexities in parameter estimation. As a future work, we highlight this possibility as a promising future direction, especially for highly non-stationary ECG segments.

While our proposed Bayesian circular SDE framework effectively models phase dynamics in ECG signals, the current study is intentionally focused on cardiac rhythm analysis. This narrow scope limits generalizability to other physiological domains such as respiratory or neural rhythms, which often exhibit more complex or multiscale oscillatory behavior.

- **Respiratory signals:** Phase modeling of breathing cycles, particularly under irregular respiration or sleep apnea conditions, where amplitude and frequency both fluctuate.
- **Neural oscillations:** Application to brain signals (e.g., EEG), capturing transient phase synchrony, entrainment, and disruptions during cognitive tasks or epileptic episodes.
- **Multimodal fusion:** Combining phase information across multiple physiological signals using a hierarchical circular Bayesian model for joint rhythm monitoring.

## 12 Conclusion

In this work, we introduced a Bayesian circular SDE framework for modeling the stochastic evolution of cardiac phase derived from ECG signals. By focusing on the phase domain rather than amplitude, our approach detects arrhythmic anomalies that are subtle and often missed by traditional linear or frequency-based methods.

Simulation experiments using phase-shifted synthetic ECG data demonstrate the utility of our approach. Compared to the AR model (F1-score: 0.309) and Fourier-based methods (F1-score: 0.070), our Bayesian circular SDE model achieved higher accuracy (91.25%) and showed competitive or superior performance in scenarios involving phase-only anomalies. Additionally, we highlighted the model's short-term predictive capability and visualized results through rose plots and circular histograms.

The study primarily addresses ECG signals and assumes high-quality phase extraction via the Hilbert transform. It does not explicitly model signal noise or handle multiscale phase variability seen in neural or respiratory data.

We aim to extend the framework to respiratory and neural signals, where circular phase dynamics also play a key role. Hierarchical modeling of joint circular rhythms, robust inference in noisy conditions, and real-time detection using sequential Bayesian updating (e.g., particle filters) are promising directions for clinical application.

Overall, our work lays the theoretical and computational foundation for the application of circular statistics in biomedical signal processing—filling an important gap in the existing literature.

## Supporting information

**This paper's supplementary material, contained Mathematical Preliminaries, modeling framework, etc.** These materials offer essential context and details supporting the main manuscript.
(PDF)

## Author contributions

**Conceptualization:** Debashis Chatterjee, Subhrajit Saha, Prithwish Ghosh.

**Data curation:** Debashis Chatterjee.

**Formal analysis:** Debashis Chatterjee, Subhrajit Saha, Prithwish Ghosh.

**Investigation:** Debashis Chatterjee, Prithwish Ghosh.

**Methodology:** Debashis Chatterjee, Subhrajit Saha.

**Project administration:** Debashis Chatterjee, Prithwish Ghosh.

**Resources:** Debashis Chatterjee.

**Software:** Debashis Chatterjee, Subhrajit Saha.

**Supervision:** Debashis Chatterjee.

**Validation:** Debashis Chatterjee.

**Visualization:** Debashis Chatterjee.

**Writing – original draft:** Debashis Chatterjee, Subhrajit Saha, Prithwish Ghosh.

**Writing – review & editing:** Debashis Chatterjee, Subhrajit Saha, Prithwish Ghosh.

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
