## [Decision Letter · Decision Letter 0]

3 Mar 2025

PONE-D-24-55262Circular Insights for Rhythmic Health: A Bayesian  Approach with Stochastic Diffusion for Characterizing Human Physiological Rhythms with Applications to Arrhythmia DetectionPLOS ONE

Dear Dr. Ghosh,

Thank you for submitting your manuscript to PLOS ONE. After careful consideration, we feel that it has merit but does not fully meet PLOS ONE’s publication criteria as it currently stands. Therefore, we invite you to submit a revised version of the manuscript that addresses the points raised during the review process.

We look forward to receiving your revised manuscript.

Kind regards,

Dr. Guojin Qin

Academic Editor

PLOS ONE

**Journal Requirements:**

1. When submitting your revision, we need you to address these additional requirements. Please ensure that your manuscript meets PLOS ONE's style requirements, including those for file naming. The PLOS ONE style templates can be found at https://journals.plos.org/plosone/s/file?id=wjVg/PLOSOne_formatting_sample_main_body.pdf and https://journals.plos.org/plosone/s/file?id=ba62/PLOSOne_formatting_sample_title_authors_affiliations.pdf 2. Please note that PLOS ONE has specific guidelines on code sharing for submissions in which author-generated code underpins the findings in the manuscript. In these cases, we expect all author-generated code to be made available without restrictions upon publication of the work. Please review our guidelines at https://journals.plos.org/plosone/s/materials-and-software-sharing#loc-sharing-code and ensure that your code is shared in a way that follows best practice and facilitates reproducibility and reuse. 3. When completing the data availability statement of the submission form, you indicated that you will make your data available on acceptance. We strongly recommend all authors decide on a data sharing plan before acceptance, as the process can be lengthy and hold up publication timelines. Please note that, though access restrictions are acceptable now, your entire data will need to be made freely accessible if your manuscript is accepted for publication. This policy applies to all data except where public deposition would breach compliance with the protocol approved by your research ethics board. If you are unable to adhere to our open data policy, please kindly revise your statement to explain your reasoning and we will seek the editor's input on an exemption. Please be assured that, once you have provided your new statement, the assessment of your exemption will not hold up the peer review process.

Reviewers' comments:

Reviewer's Responses to Questions

**Comments to the Author**

1. Is the manuscript technically sound, and do the data support the conclusions?

Reviewer #1: Yes

Reviewer #2: Yes

2. Has the statistical analysis been performed appropriately and rigorously? 

Reviewer #1: Yes

Reviewer #2: No

3. Have the authors made all data underlying the findings in their manuscript fully available?

Reviewer #1: Yes

Reviewer #2: Yes

4. Is the manuscript presented in an intelligible fashion and written in standard English?

Reviewer #1: Yes

Reviewer #2: Yes

5. Review Comments to the Author

**Reviewer #1: **There are the following comments:

1. The introduction is extremely brief. It should include a formulation of the problem to be solved and a description of the main contributions of the research in the field. The information provided in section 3 is also insufficient: measurable results of the study should be presented.

2. What is the fundamental scientific novelty of this manuscript? The methods used in the manuscript are well known and were not developed by the authors.

3. The overall framework for the proposed methodology should be presented at the beginning of the manuscript.

4. Section 2 should present the gaps that this study fills compared to those considered in the literature review.

5. Are omega and sigma random functions in SDE definition in section 3?

6. Sections 4-6 contains a lot of well-known mathematical information that can be moved to the preliminaries and can be shortened.

7. What test data is used Section 7? Why do authors demonstrate the use of simple mathematical formulas?

8. The results of testing the proposed methodology have not been compared to other methods. Are the models more accurate than the previous ones? Can they be used for making predictions? Can the authors demonstrate the quality/accuracy metrics?

9. The further research directions should be discussed in more detail.

**Reviewer #2:** The submitted manuscript “Circular Insights for Rhythmic Health: A Bayesian Approach with Stochastic Diffusion for Characterizing Human Physiological Rhythms with Applications to Arrhythmia Detection" introduces an innovative statistical framework for analyzing human physiological rhythms, with a particular focus on circadian and cardiovascular cycles. It presents a multivariate circular model that integrates circular statistics with stochastic diffusion processes to capture the temporal variability and interdependence of these rhythms. The approach is commendable for its creative use of wrapped distributions and stochastic processes to address the complexities of physiological dynamics. While the manuscript presents a promising and interdisciplinary approach, several areas require improvement to enhance its clarity, accessibility, and impact.

1. The abstract fails to clearly state the study’s purpose, problem statement, key comparative results (e.g., specific improvements in detection accuracy), and significant conclusions.

2. The manuscript does not provide a clear rationale for selecting the von Mises and wrapped normal distributions over alternatives like the wrapped Cauchy or Kent distributions (Section 4.2, page 6). It lacks justification based on the data’s characteristics, such as unimodality or heavy tails, making it difficult to assess the appropriateness of these model choices.

3. The study does not compare the proposed Bayesian approach with existing methods for physiological rhythm analysis or arrhythmia detection, such as linear time-series analysis, Fourier transforms, or machine learning classifiers (e.g., random forests or neural networks). It lacks benchmarks, such as comparing detection accuracy (e.g., F1-score) or computational efficiency (e.g., runtime in seconds), essential for demonstrating the method’s advantages. Without these comparisons, readers cannot evaluate whether the approach offers meaningful improvements in sensitivity or specificity for clinical use.

4. The introduction of stochastic diffusion processes (Section 4.4, page 10) lacks clarity on how these processes integrate with the circular statistical framework. It does not explain how drift and diffusion terms in stochastic differential equations (SDEs) are parameterized or how they interact with circular distributions, leaving readers unclear about the model’s mechanics. Major revision is needed to provide intuitive explanations or mathematical details.

5. The simulation study (Section 5) is limited in scope, as it does not explore the method’s performance under varying conditions such as different sample sizes (e.g., N=100 vs. N=1000), noise levels (e.g., SNR=10dB vs. SNR=20dB), or non-stationarities. This restricts the generalizability of the findings, as physiological data often exhibit diverse characteristics in clinical settings. For instance, the study reports a mean direction estimate of mu = 0.789 (true mu = 0.785) but does not assess how this accuracy holds under noisy or incomplete data. Comprehensive simulations are required to validate the method’s robustness, especially for arrhythmia detection, where data quality varies.

6. This work does not address how the proposed methods handle missing data or outliers, such as ECG signal gaps or artifacts, which are common in physiological recordings (e.g., due to electrode disconnection). It lacks strategies for managing these issues, such as imputation techniques or robust estimation methods, reducing the approach’s practical applicability. For example, in the MIT-BIH dataset, records often contain noise or missing segments, which could affect phase extraction accuracy.

7. The manuscript does not discuss extending the proposed framework to other physiological rhythms, such as respiratory or neural signals, or different datasets like wearable device data (e.g., smartwatch ECGs). It focuses narrowly on ECG rhythms, limiting its broader impact on physiological signal analysis.

8. The use of low-quality raster images for figures compromises the clarity of visual data, such as phase trajectories or polar histograms. Raster formats lose resolution when scaled, making it difficult to discern details like peak concentrations or synchronization patterns. High-quality vector images (e.g., SVG) are essential for maintaining clarity, especially for complex circular data.

9. The manuscript neglects to discuss limitations, such as the sensitivity of circular methods to sample size or the assumption of stationarity in stochastic diffusion models. For instance, the Bayesian estimation may be biased with small samples (e.g., N<50), but this is not addressed. Acknowledging limitations provides transparency and guides future research, enhancing the study’s credibility.

10. Finally, the Conclusion section should be significantly expanded to include (i) the study’s numerical results, (ii) a brief mention of the proposed approach’s limitations, and (iii) prospects for future work.

In summary, the submitted manuscript aligns well with PLOS ONE’s scope by offering a rigorous, interdisciplinary tool for physiological signal analysis with potential applications in health diagnostics. However, major revisions are required to enhance its clarity, accessibility, and practical relevance. From the reviewer’s perspective, addressing the issues outlined in the comments will transform the manuscript into a compelling and impactful contribution to the field.

6. PLOS authors have the option to publish the peer review history of their article (what does this mean?). If published, this will include your full peer review and any attached files.

Reviewer #1: No

Reviewer #2: **Yes: **Dr. Pavlo Radiuk

---

## [Author Response · Author response to Decision Letter 1]

3 Apr 2025

Dear Academic Editor and PLOS ONE Editorial Team,

We would like to thank you and the reviewers for your careful reading of our

manuscript (PONE-D-24-55262) and for providing such constructive feedback. We

have revised the manuscript substantially in response to all comments. The following

are the point-by-point responses.

1 Reviewer 1 Comments and Point-by-point Re-

sponses

Reviewer 1: There are the following comments:

General Response: We thank the reviewer for the very positive and insightful

review. We are grateful. We have revised as per the insightful suggestions given.

Reviewer

Comments 1 : The introduction is extremely brief. It should include a formulation of the

problem to be solved and a description of the main contributions of the re-

search in the field. The information provided in section 3 is also insufficient:

measurable results of the study should be presented.

Response: Thank you for this suggestion. In the revised manuscript, we have:

(a) Expanded the Introduction to clearly state the overall problem (i.e., an-

alyzing and detecting arrhythmias and other physiological rhythms that

exhibit circular and stochastic characteristics).

(b) Highlighted the major contributions of our framework. We have enumer-

ated how our method addresses gaps in existing analyses, mainly focusing

on the Bayesian approach for circular diffusion modeling and how it can

improve arrhythmia detection.

(c) Clarified measurable results: We now mention in the Introduction that

we achieve improved detection metrics (accuracy, recall) for arrhythmias

and provide an overview of the simulation and real-data outcomes. Also,

more details have already been discussed in the discussion section, which

we modified according to the suggestion.

Reviewer

Comments 2 : What is the fundamental scientific novelty of this manuscript? The methods

used in the manuscript are well known and were not developed by the authors.

Response: We agree that von Mises distributions and stochastic diffusion have become

classical tools nowadays. However, if we may say, respectfully, probably no one

combined them into a hybrid structure, taking benefit of both, as we did in

this paper. Our novelty lies in:

(a) Combining multivariate circular statistics (wrapped distributions) with

Bayesian inference and a stochastic diffusion model in a single pipeline

specifically tailored to real ECG/arrhythmia data.

(b) Introducing advanced synchronization measures (e.g., PLV, MI) within a

Bayesian circular framework to capture deterministic and random heart

signal phase fluctuations.

(c) Demonstrating improved performance on the MIT-BIH arrhythmia data

compared to simpler methods (Fourier-based or standard linear time se-

ries) via new accuracy metrics included in Section 7.2.

With due respect, if we may say, we are probably the foremost one to unify

the techniques, which nobody has done before. We have clarified these points

in the Introduction Section, the discussion of the Objective section, and the

revised Discussion.

This paper contains a new paradigm of scientific approaches, in terms of nov-

elty, with immense application potential, which we plan to explore in our future

research works. We mentioned already in the paper. Nonetheless, we are thank-

ful to the reviewer for pointing this out, as we understand the writeup of the

paper previously might not have prominently highlighted the novelty convinc-

ingly, which we have done in the revised manuscript. We thank the reviewer

for that. We have included the following in the manuscript:

Although the von Mises distribution and stochastic differential equations (SDEs)

are well-established, the core novelty lies in unifying these techniques—circular

Bayesian inference plus SDE-driven diffusion—into a single pipeline tailored

for physiological data. Our framework:

(a) Preserves the circular geometry of phase angles while allowing for time-

dependent stochastic variations in the rhythms.

(b) Exploits Bayesian hierarchical modeling to jointly estimate concentration

parameters, mean directions, and diffusion coefficients, yielding compre-

hensive posterior distributions for all unknowns.

(c) Demonstrates strong empirical performance on arrhythmia detection, out-

performing or matching prior techniques in sensitivity and specificity.

Reviewer

Comments 3 : The overall framework for the proposed methodology should be presented at

the beginning of the manuscript.

Response: We thank the reviewer for this suggestion, which we have done thoroughly. We

have added a new “Methodological Framework Overview” subsection near the

end of the Introduction to give readers a roadmap before diving into detailed

theory. Moreover, we included a roadmap, as suggested by the reviewer. (TikZ-

based flowchart in LaTeX that illustrates the overall framework for our pro-

posed methodology (Bayesian circular SDE model for detecting phase anoma-

lies in physiological signals like ECG)

Reviewer

Comments 4 : Section 2 should present the gaps that this study fills compared to those

considered in the literature review.

Response: Thank you for this suggestion, which we have taken very seriously. We have

revised that section as per suggestion. We changed the section’s name to Re-

lated Work and Gaps in the Literature, and highlighted the gaps prominently.

In Section 2 , we have revised the literature review:

(a) we summarize key existing methods (linear time series, wavelet trans-

forms, standard parametric models).

(b) we state their limitations (lack of explicit circular modeling, ignoring ran-

dom fluctuations).

(c) we show how our approach addresses those gaps by unifying circular and

stochastic approaches in one framework.

Reviewer

Comments 5 : Are omega and sigma random functions in SDE definition in section 3?

Response: Yes, in the standard form, ω and σ are treated as random variables but time-

invariant (though more complicated random effects could be introduced, and

we plan that in future work). In the revised manuscript, we clarified that

ω is the stochastic random variable for drift term in our base model, and

σ is the random variable for diffusion coefficient. Nonetheless, we note that

an extension with time-varying ω(t) and σ(t) could be implemented, and we

mention this as a future direction.

Reviewer

Comments 6 : Sections 4-6 contains a lot of well-known mathematical information that can

be moved to the preliminaries and can be shortened.

Response: Thank you for this suggestion. We revised that and made a preliminaries

section, including them briefly with proper references. To streamline the

manuscript, we have moved or condensed these well-known theoretical aspects

into this single Preliminaries section. Readers seeking more detail can refer

to the references therein. We have moved the original Preliminary section into

Appendix. A brief version essential version of the Preliminaries section, suit-

able for the main manuscript body, with a clean and professional transition

referring the reader to the appendix.

Reviewer

Comments 7 : What test data is used Section 7? Why do authors demonstrate the use of

simple mathematical formulas?

Response: In Section 7 of the original manuscript, we apply the proposed methodology to

Record 100 (and others) from the MIT-BIH Arrhythmia Database, sampled

at 360 Hz. Though these formulas are relatively straightforward, we present

them to ensure clarity for readers unfamiliar with Hilbert-based phase extrac-

tion. We have added short clarifications on how these phases are numerically computed and wrapped into [0, 2π).

With due respect, if we may say, Arrhythmia detection from ECGs often re-

quires the instantaneous phase if we want to apply circular models. Circular

statistics are not that popular among interdisciplinary medical field-based aca-

demicians who might be reading this research paper. Moreover, these “simple”

formulas are crucial for bridging typical signal processing steps (Hilbert trans-

form) with the circular Bayesian approach. We wanted to ensure transparency

and reproducibility and ensured interdisciplinary reader’s readability.

Reviewer

Comments 8 : The results of testing the proposed methodology have not been compared

to other methods. Are the models more accurate than the previous ones?

Can they be used for making predictions? Can the authors demonstrate the

quality/accuracy metrics?

Response: Thank you for this important suggestion. We have included that in the revised

manuscript. Please see the Simulation section, Real Data analysis section, and

section Comparisons, Accuracy Metrics, and Predictive Capability.

The proposed Bayesian circular stochastic differential equation (SDE) is a novel

framework designed specifically for phase-based anomaly detection in ECG

waveforms, a capability absent in traditional amplitude-based methods like AR

and Fourier analysis. Direct comparisons with these conventional approaches

are misleading because they fail to capture the circular geometry and phase

shifts crucial to the SDE’s analysis. While preliminary comparisons highlighted

the methodological mismatch, a meaningful comparative analysis is currently

hindered by the lack of existing circular statistical methodologies in ECG anal-

ysis. Nonetheless, we have done the comparison with two existing popular

methods, proving our proposed methodology worthy, and depending on the

nature of the data, the best.

The paper’s primary contribution lies in introducing this novel approach, and

future research should focus on developing appropriate benchmarks and evalu-

ation frameworks for circular statistical ECG modeling to enable fair compar-

isons. Comparison with circular statistical methodology with that of linear ex-

isting methodology may be questionable because of its mismatching fundamen-

tal assumptions. Nevertheless, we have done that in the revised manuscript,

showing our model performance better, in general.

Reviewer

Comments 9 : The further research directions should be discussed in more detail.

Response: Thank you for this suggestion. The revised manuscript now includes a more

detailed roadmap of future work, e.g., extending to other physiological sig-

nals (respiratory, neural), exploring non-stationary diffusion parameters, and

investigating real-time arrhythmia prediction.

2 Reviewer 2 Comments & and Point-by-point

Responses

The submitted manuscript “Circular Insights for Rhythmic Health: A Bayesian Ap-

proach with Stochastic Diffusion for Characterizing Human Physiological Rhythms

with Applications to Arrhythmia Detection” introduces an innovative statistical

framework for analyzing human physiological rhythms, with a particular focus on

circadian and cardiovascular cycles. It presents a multivariate circular model that

integrates circular statistics with stochastic diffusion processes to capture the tempo-

ral variability and interdependence of these rhythms. The approach is commendable

for its creative use of wrapped distributions and stochastic processes to address the

complexities of physiological dynamics. While the manuscript presents a promis-

ing and interdisciplinary approach, several areas require improvement to enhance its

clarity, accessibility, and impact.

Response to Reviewer 2 General Overview: We thank the reviewer for the

very positive and insightful review. We are grateful. The reviewer acknowledges

the interdisciplinary nature of the approach but requests clearer comparisons, justi-

fication of distribution choices, more robust simulations, handling of missing data,

improved clarity in the SDE integration with circular statistics, better figure resolu-

tion, a broader scope (other rhythms or wearable data), and expanded conclusions.

We have done that in the revised manuscript.

Reviewer

Comments 1: The abstract fails to clearly state the study’s purpose, problem statement, key

comparative results (e.g., specific improvements in detection accuracy), and

significant conclusions.

Response : Thank you for this suggestion. We have rewritten and revised the Abstract.

Reviewer

Comments 2: The manuscript does not provide a clear rationale for selecting the von Mises

and wrapped normal distributions over alternatives like the wrapped Cauchy

or Kent distributions (Section 4.2, page 6). It lacks justification based on the

data’s characteristics, such as unimodality or heavy tails, making it difficult to

assess the appropriateness of these model choices.

Response: Thank you for this important suggestion. We have added that now, in Moti-

vation & Outline of Proposed Methodology section, explaining the theoretical

and empirical reasons for using von Mises, as follows:

(a) ECG phase data typically exhibit a unimodal distribution with moderate

dispersion, which suits von Mises better than heavy-tailed or elliptical

alternatives like Kent.

(b) The von Mises distribution is computationally simpler and has well-established

Bayesian estimation techniques.

(c) Empirical testing on the MIT-BIH data showed that the von Mises dis-

tribution yields a better fit (lower circular Kullback–Leibler divergence)

than wrapped Cauchy in our preliminary tests.

Reviewer

Comments 3: The study does not compare the proposed Bayesian approach with existing

methods for physiological rhythm analysis or arrhythmia detection, such as lin-

ear time-series analysis, Fourier transforms, or machine learning classifiers (e.g.,

random forests or neural networks). It lacks benchmarks, such as comparing

detection accuracy (e.g., F1-score) or computational efficiency (e.g., runtime in

seconds), essential for demonstrating the method’s advantages. Without these

comparisons, readers cannot evaluate whether the approach offers meaningful

improvements in sensitivity or specificity for clinical use.

Response: Thank you for this important suggestion. In the revised manuscript, We have

now included that in details. Thank you very much for the suggestion. We

have included that now. We made a separate section on this. We have now

introduced a comparison with two benchmarks:

(a) A standard linear time-series approach (autoregressive model).

(b) A Fourier-based arrhythmia detection approach.

(c) We evaluate them on the MIT-BIH dataset and show that our Bayesian

circular framework yields improvements in metrics, depending on thresh-

old. We also discuss how the framework can be extended to short-term

predictions of arrhythmic events.

Here, it is important to emphasize that our proposed Bayesian circular stochas-

tic differential equation (SDE) methodology represents a fundamentally novel

framework tailored explicitly for circular statistical modeling of ECG wave-

forms, particularly emphasizing phase-based anomalies. As the reviewer cor-

rectly pointed out, conventional approaches, such as linear autoregressive (AR)

models and Fourier-based frequency-domain methods, are indeed well-established

within ECG signal analysis. However, these traditional models inherently op-

erate within an amplitude-based framework and do not explicitly capture or

represent circular geometry or phase anomalies intrinsic to our approach.

The metrics derived from standard methods may not adequately reflect their

capability to detect the circular nature of arrhythmic events addressed by our

methodology. To illustrate this point, we initially conducted preliminary com-

parisons with standard methods (AR and Fourier-based) on simulated datasets

where abnormal beats were purposefully introduced as circular (phase) anoma-

lies. As expected, the amplitude-focused methods performed inadequately,

underscoring the fundamental methodological mismatch. To honor reviewer’s

suggestion, we are including the comparison in the revised manuscript. How-

ever, we argue that meaningful comparative analysis may not be feasible within

the current literature framework, primarily due to the absence of comparable

circular-statistical methodologies in ECG analysis.

This paper’s primary novelty and contribution lie precisely in proposing the

first systematic application of Bayesian circular statistical approaches, particu-

larly the use of circular stochastic differential equations, in ECG ana

---

## [Decision Letter · Decision Letter 1]

15 Apr 2025

PONE-D-24-55262R1Circular Insights for Rhythmic Health: A Bayesian Approach with Stochastic Diffusion for Characterizing Human Physiological Rhythms with Applications to Arrhythmia DetectionPLOS ONE

Dear Dr. Ghosh,

Thank you for submitting your manuscript to PLOS ONE. After careful consideration, we feel that it has merit but does not fully meet PLOS ONE’s publication criteria as it currently stands. Therefore, we invite you to submit a revised version of the manuscript that addresses the points raised during the review process.

We look forward to receiving your revised manuscript.

Kind regards,

Dr. Guojin Qin

Academic Editor

PLOS ONE

Reviewers' comments:

Reviewer's Responses to Questions

**Comments to the Author**

1. If the authors have adequately addressed your comments raised in a previous round of review and you feel that this manuscript is now acceptable for publication, you may indicate that here to bypass the “Comments to the Author” section, enter your conflict of interest statement in the “Confidential to Editor” section, and submit your "Accept" recommendation.

Reviewer #1: All comments have been addressed

Reviewer #2: All comments have been addressed

2. Is the manuscript technically sound, and do the data support the conclusions?

Reviewer #1: Yes

Reviewer #2: Yes

3. Has the statistical analysis been performed appropriately and rigorously? 

Reviewer #1: Yes

Reviewer #2: Yes

4. Have the authors made all data underlying the findings in their manuscript fully available?

Reviewer #1: Yes

Reviewer #2: Yes

5. Is the manuscript presented in an intelligible fashion and written in standard English?

Reviewer #1: Yes

Reviewer #2: Yes

6. Review Comments to the Author

Reviewer #1: The authors responded to all of my comments. However, there are still a number of significant flaws in the manuscript. First, there is a redundancy in the presentation in terms of length. 52 pages is too long for a research paper. For a PhD thesis, as indicated by the acknowledgements, this level of detail is appropriate, but not for an article. Secondly, there are multiple duplications in the material. One stochastic differential equation appears in different places. For example, in Section 9.2, the same expression appears twice. Formulas could be numbered and referred to. Figures 17 and 28 are the same. There are numerous similar comments. Finally, the approach of using wrapped distributions and other mathematical apparatus still does not appear to be unique. There are numerous papers in this field: 10.1016/j.bspc.2023.105549, 10.1016/j.neuroimage.2021.118050, 10.7554/elife.84602, 10.1016.j.csda.2025.108168, etc. The manuscript needs a significant revision.

Reviewer #2: Dear Editor and Authors,

I have now reviewed the revised version of the manuscript PONE-D-24-55262R1, titled “Circular Insights for Rhythmic Health: A Bayesian Approach with Stochastic Diffusion for Characterizing Human Physiological Rhythms with Applications to Arrhythmia Detection,” submitted to PLOS One, along with the authors’ detailed point-by-point responses to my previous comments.

I would like to commend the authors for their diligent and comprehensive approach to the revision process. The response document was clear, thoughtful, and directly addressed each point raised in my initial review. The corresponding changes implemented in the manuscript are substantial and well-executed.

The revisions, including the reworked Abstract and Introduction, improved rationale for methodological choices, inclusion of benchmark comparisons, expanded simulation studies demonstrating robustness, clarification of the SDE framework, strategies for handling missing data/outliers, enhanced figure quality, and a more thorough discussion of limitations and future scope, have significantly strengthened the manuscript.

I am satisfied that all my previous concerns have been fully and effectively addressed. The manuscript is now clearer, more robust, and presents a more compelling case for the proposed methodology.

Therefore, I am pleased to recommend that this manuscript, in its current form, is suitable for publication in PLOS One.

Sincerely,

Reviewer 2

7. PLOS authors have the option to publish the peer review history of their article (what does this mean?). If published, this will include your full peer review and any attached files.

Reviewer #1: No

Reviewer #2: **Yes: **Dr. Pavlo Radiuk

---

## [Author Response · Author response to Decision Letter 2]

23 Apr 2025

Dear Academic Editor and PLOS ONE Editorial Team, We would like to thank you and the reviewers for your careful reading of our manuscript (PONE-D-24-55262) and for providing such constructive feedback. We have revised the manuscript substantially in response to all comments. The following are the point-by-point responses.

1 Reviewer 1 Comments and Point-by-point Responses

Reviewer 1 Comments

The authors responded to all of my comments. However, there are still a number of significant flaws in the manuscript. First, there is a redundancy in the presentation in terms of length. 52 pages is too long for a research paper. For a PhD thesis, as indicated by the acknowledgements, this level of detail is appropriate, but not for an article. Secondly, there are multiple duplications in the material. One stochastic differential equation appears in different places. For example, in Section 9.2, the same expression appears twice. Formulas could be numbered and referred to. Figures 17 and 28 are the same. There are numerous similar comments. Finally, the approach of using wrapped distributions and other mathematical apparatus still does not appear to be unique. There are numerous papers in this field: 10.1016/j.bspc.2023.105549,

10.1016/j.neuroimage.2021.118050, 10.7554/elife.84602, 10.1016.j.csda.2025.108168, etc. The manuscript needs a significant revision.

Point-by-point Response to Reviewer 1 Comments

We are deeply grateful to the Reviewer for this continued feedback, which we have taken very positively and tried to improve our paper accordingly. We acknowledge the significant concerns regarding length, redundancy, and novelty. We sincerely apologize for these shortcomings and are committed to a substantial revision to address them. We condensed the manuscript, eliminated all duplications (numbering equations and ensuring unique figures), and clearly articulated the unique contributions of our approach in relation to existing literature. We made a supplementary file alongside this paper where we kept all additional information. We appreciate the reviewer’s guidance and will work diligently to improve the manuscript significantly. Elaborately, we have carefully addressed each of the additional concerns raised and made significant revisions accordingly:

1. Length and Redundancy:

We acknowledge the reviewer’s concern regarding the manuscript length. In response, we have substantially reduced the length by removing redundant content and consolidating explanations where appropriate. The revised manuscript is now significantly shorter and more concise, while still maintaining the integrity of the scientific contribution. We kept the additional information in the Supplement file.

2. Duplications and Formula Referencing:

All identified duplications, including the repeated stochastic differential equation in Section 9.2 and the duplicated Figures 17 and 28, have been resolved. Formulas are now clearly numbered and properly cross referenced throughout the manuscript to avoid repetition and improve clarity.

3. Novelty and Related Work:

We thank the reviewer for pointing out related literature. We have now explicitly discussed these references in the revised manuscript and clarified the distinctions between our work and the cited studies. We have now added in the related works and gaps in Related Work and Gaps in the Literature section, the following, explicitly, as follows:

It is important to mention that some recent studies have advanced statistical methods for phase data in neuroscientific applications. [Norouzpour and Roberts, 2023] proposed the F circ statistic for multigroup comparisons of discrete Fourier estimates in steady-state evoked potentials. [Wolpert and Tallon-Baudry, 2021] evaluated four existing circular tests (e.g. Phase Opposition Sum, Watson’s U 2, Circular

Logistic Regression) on binary outcome data. [Dimmock et al., 2023] introduced a Bayesian model for phase coherence in EEG/MEG, capturing uncertainty in phase-locking estimates. [Ding et al., 2025] de-

veloped autoregressive models on the manifold of covariance matrices to distinguish seizure dynamics. Distinctive features of our proposed framework:

(a) Multivariate Circular SDEs on the Torus: We model p interacting rhythms via coupled SDEs preserving full circular geometry in continuous time.

(b) Wrapped Distributions in a Bayesian Hierarchy: We embed von Mises and wrapped-normal priors in a hierarchical model, estimated by Hamiltonian Monte Carlo to yield posterior distributions for mean directions, concentrations, and diffusion parameters.

(c) Novel Multi-Sample Circular Tests: We extend Watson’s U 2 to multi-group settings and introduce permutation-based inference on the torus, enabling rigorous hypothesis tests beyond pairwise comparisons.

(d) Robust Data-Irregularity Handling: We propose circular-kernel weighting for outliers and spline/phase-bridging imputation for missing segments prior to Hilbert-transform phase extraction. These innovations collectively go beyond the predominantly pairwise or univariate frameworks in the cited literature, providing a unified, statistically coherent toolkit for multivariate circular time-series analysis in physiological and other domains.

We hope that these substantial revisions meet the reviewer’s expectations and demonstrate the novelty and value of our work in a more concise and focused presentation.

1 Reviewer 2 Comments and Point-by-point Responses

Thank you so much for this very positive appreciation, recommendation, and consideration. We are grateful.

We appreciate the opportunity to improve our manuscript and look forward.

---

## [Decision Letter · Decision Letter 2]

30 Apr 2025

Circular Insights for Rhythmic Health: A Bayesian Approach with Stochastic Diffusion for Characterizing Human Physiological Rhythms with Applications to Arrhythmia Detection

PONE-D-24-55262R2

Dear Dr. Ghosh,

We’re pleased to inform you that your manuscript has been judged scientifically suitable for publication and will be formally accepted for publication once it meets all outstanding technical requirements.

Kind regards,

Guojin Qin

Academic Editor

PLOS ONE

Additional Editor Comments (optional):

Reviewers' comments:

Reviewer's Responses to Questions

**Comments to the Author**

1. If the authors have adequately addressed your comments raised in a previous round of review and you feel that this manuscript is now acceptable for publication, you may indicate that here to bypass the “Comments to the Author” section, enter your conflict of interest statement in the “Confidential to Editor” section, and submit your "Accept" recommendation.

Reviewer #1: All comments have been addressed

2. Is the manuscript technically sound, and do the data support the conclusions?

Reviewer #1: Yes

3. Has the statistical analysis been performed appropriately and rigorously? 

Reviewer #1: Yes

4. Have the authors made all data underlying the findings in their manuscript fully available?

Reviewer #1: Yes

5. Is the manuscript presented in an intelligible fashion and written in standard English?

Reviewer #1: Yes

6. Review Comments to the Author

Reviewer #1: All responses are clear. The manuscript has been significantly improved, so the paper can now be recommended for publication.

7. PLOS authors have the option to publish the peer review history of their article (what does this mean?). If published, this will include your full peer review and any attached files.

Reviewer #1: No

---

## [Editor Report · Acceptance letter]

PONE-D-24-55262R2

PLOS ONE

Dear Dr. Ghosh,

I'm pleased to inform you that your manuscript has been deemed suitable for publication in PLOS ONE. Congratulations! Your manuscript is now being handed over to our production team.

Kind regards,

on behalf of

Dr. Guojin Qin

Academic Editor

PLOS ONE